# AUGMENTING NEGATIVE REPRESENTATIONS FOR CONTINUAL SELF-SUPERVISED LEARNING

## ABSTRACT

We introduce a novel and general loss function, called Augmented Negatives (AugNeg), for effective continual self-supervised learning (CSSL). We first argue that the conventional loss form of continual learning which consists of single task-specific loss (for plasticity) and a regularizer (for stability) may not be ideal for contrastive loss based CSSL that focus on representation learning. Our reasoning is that, in contrastive learning based methods, the task-specific loss would suffer from decreasing diversity of negative samples and the regularizer may hinder learning new distinctive representations. To that end, we propose AugNeg that consists of two losses with symmetric dependence on current and past models' negative representations. We argue our model can naturally find good trade-off between the plasticity and stability without any explicit hyperparameter tuning. Furthermore, we present that the idea of utilizing augmented negatives can be applied to CSSL with non-contrastive learning by adding an additional regularization term. We validate the effectiveness of our approach through extensive experiments, demonstrating that applying the AugNeg loss achieves superior performance compared to other state-of-the-art methods, in both contrastive and non-contrastive learning algorithms.

## 1 INTRODUCTION

Self-supervised learning (SSL) has recently emerged as a cost-efficient approach for training neural networks, eliminating the need for laborious data labelling (Gui et al., 2023). Specifically, the representations learned by recent SSL methods (*e.g.,* MoCo (He et al., 2020), SimCLR (Chen et al., 2020a), BarlowTwins (Zbontar et al., 2021), BYOL (Grill et al., 2020), and VICReg (Bardes et al., 2022) are shown to have excellent quality, comparable to those learned from supervised learning. Despite such success, huge memory and computational complexities are the apparent bottlenecks for easily maintaining and updating the self-supervised learned models, since they typically require large-scale unsupervised data, large mini-batch sizes, and numerous gradient update steps for training.

To that end, continual self-supervised learning (CSSL), in which the aim is to learn progressively improved representations from a sequence of unsupervised data, can be an efficient alternative to the high-cost, jointly trained self-supervised learning. With such motivation, several recent studies (Madaan et al., 2022; Hu et al., 2022; Fini et al., 2022) have considered the contrastive learning based CSSL and showed their effectiveness in maintaining representation continuity. Despite the positive results, we note that the core idea for those methods is mainly borrowed from the large body of continual learning research for supervised learning (Parisi et al., 2019; Delange et al., 2021; Wang et al., 2023). Namely, a typical supervised continual learning method can be generally described as employing a single-task loss term for the new task (*e.g.,* cross-entropy or supervised contrastive loss (Khosla et al., 2020)) together with a certain type of regularization (*e.g.,* distillation-based (Li & Hoiem, 2017; Douillard et al., 2020; Kang et al., 2022; Wang et al., 2022; Cha et al., 2021a) or norm-based (Kirkpatrick et al., 2017; Aljundi et al., 2018; Jung et al., 2020; Ahn et al., 2019; Cha et al., 2021b) or replay-sample based terms (Wu et al., 2019; Rebuffi et al., 2017)) to prevent forgetting; the recent state-of-the-art CSSL methods simply follow that approach with *unsupervised self-supervised* loss terms.

In this regard, we raise two issues on the current self-supervised loss based CSSL approach. Firstly, it is not clear whether adding a regularization term to the existing self-supervised loss is the best way to achieve successful CSSL. Namely, typical regularization terms are essentially designed to

maintain the representations of previous model, but they may hinder the capability of learning better representations while learning from new task data (Cha et al., 2023). Secondly, the current approach would suffer from the decrease in diversity of negative samples when the contrastive loss is used as the single-task loss, since the negative samples are only mined from the current task data. Several studies have shown that the diversity of negative samples is crucial for learning good representations for contrastive learning (Wang & Isola, 2020; Chen et al., 2020a; Tao et al., 2022), and accordingly, the current CSSL methods may be inherently limited in learning good representations compared to the jointly trained SSL models which have access to full data.

To address above limitations, we propose an effective method for augmenting negative representations for CSSL. First, we consider the case of using InfoNCE -type *contrastive* loss (Oord et al., 2018), *e.g.,* SimCLR or MoCo, which explicitly utilizes the negative samples while learning. Namely, we propose novel loss functions, dubbed as **AugNeg**, by modifying the ordinary InfoNCE loss as well as the regularization term for the contrastive distillation (Fang et al., 2021; Tian et al., 2019) so that the former also considers the negative sample representations from the *previous* model and the latter also includes the negative samples representations from the *current* model. We argue that enhancing negative representations in this manner facilitates the CSSL methods in learning more distinct representations from the new data. It ensures that the newly acquired representations do not overlap with the previously learned ones, thereby enhancing *plasticity*. Additionally, it allows for the effective distillation of prior knowledge into the current model without interfering with the representations already learned by the current model, thus improving overall *stability*. Second, we extend above idea to the CSSL using *non-contrastive* learning methods *e.g.,* BYOL, BarlowTwins and VICReg. Namely, while those methods do not explicitly use the negative samples in the original implementations, we show that the idea of augmenting the "negative" representations from the previous model, which we will precisely define later, is helpful for the several CSSL settings. Finally, through extensive experimental analyses, our proposed algorithm demonstrates not only the gradual improvement in the quality of representations during the CSSL steps but also superior results across various forms of downstream tasks.

## 2 RELATED WORK

**Self-supervised representation learning**    There have been several recent variations for Self-Supervised Learning (SSL) (Alexey et al., 2016; Doersch et al., 2015; Vincent et al., 2010; Zhang et al., 2016; Hadsell et al., 2006; Chen et al., 2020a; He et al., 2020). Among those, *contrastive loss-based* methods have emerged as the leading approach to learn discriminative representations (Hadsell et al., 2006; Oord et al., 2018), in which the representations are learned by pulling the positive pairs together and pushing the negative samples apart. Several efficient contrastive learning methods, like MoCo (He et al., 2020; Chen et al., 2020b), SimCLR (Chen et al., 2020a), have been proposed build on the InfoNCE loss (Oord et al., 2018). Additionally, *non-contrastive learning* methods, such as Barlow Twins (Zbontar et al., 2021), BYOL (Grill et al., 2020) and VICReg (Bardes et al., 2022), have been demonstrated to yield high-quality learned representations without negatives.

**Continual learning**    Continual learning (CL) is the process of acquiring new knowledge while retaining previously learned knowledge (Parisi et al., 2019; Masana et al., 2020) from a sequence of tasks. To balance the trade-off between *plasticity*, the ability to learn new tasks well, and *stability*, the ability of retaining knowledge of previous tasks (Mermillod et al., 2013), the supervised CL research has been proposed in three categories (Parisi et al., 2019; Wang et al., 2023; Delange et al., 2021). Recently, there has been a growing interest in Continual Self-Supervised Learning (CSSL), as evidenced by several related researches (Rao et al., 2019; Madaan et al., 2022; Hu et al., 2022; Fini et al., 2022). Madaan et al. (2022) first demonstrates that CSSL can outperform supervised CL algorithms in the task-incremental learning scenario. Additionally, they show that CSSL is less prone to catastrophic forgetting due to achieving wider local minima. Another study (Hu et al., 2022) focuses on the benefits of CSSL in large-scale datasets (*e.g.*, ImageNet), demonstrating that a competitive pre-trained model can be obtained through CSSL. Lastly, the most recent work  (Fini et al., 2022) proposed a novel regularization (dubbed as CaSSLe) which effectively overcomes catastrophic forgetting in CSSL, achieving state-of-the-art performance in various scenarios of CSSL.

Our paper makes distinctive contributions compared to related works. Firstly, we identify shortcomings in a *conventional loss form* for CSSL such as those majorly used in prior works like CaSSLe, and shed new light on an issue where the *diversity of negative samples* significantly decreases in CSSL.

Secondly, we suggest utilizing additional negatives obtained through *model-based augmentation* which is specifically tailored for CSSL. This approach sets itself apart from previous augmentation methods that primarily focused on training a single-task in SSL (Zheng et al., 2021; Bai et al., 2022).

## 3 MAIN METHOD

### 3.1 PROBLEM SETTING

**Notations and preliminaries.** We evaluate the quality of CSSL methods using the setting and data for continual supervised learning similarly as in Fini et al. (2022). Namely, let $t$ be the task index, where $t \in \{1, \ldots, T\}$, and $T$ represents the maximum number of tasks. The input data and their corresponding true labels given at the $t$-th task are denoted by $x \in \mathcal{X}_t$ and $y \in \mathcal{Y}_t$, respectively[1]. We assume each training dataset for task $t$ comprises $M$ supervised pairs, denoted as $\mathcal{D}_t = \{(x_i, y_i)\}_{i=1}^M$, in which each pair is considered to be sampled from a joint distribution $p(\mathcal{X}_t, \mathcal{Y}_t)$. Note in the case of continual supervised learning (CSL) (Delange et al., 2021; Masana et al., 2020), both inputs and the labels are used, whereas in CSSL (Fini et al., 2022; Madaan et al., 2022), only input data are utilized for training, while the true labels are used only for the evaluation of the learned representations, such as linear probing or $k$-NN evaluation (Fini et al., 2022; Cha et al., 2023). We denote $h_{\theta_t}$ as the representation encoder (with parameter $\theta_t$) learned after task $t$ by a CSSL method. To evaluate the quality of $h_{\theta_t}$ via linear probing, we consider a classifier $f_{\Theta_t} = o_{\phi_t} \circ h_{\theta_t}$, in which $\Theta_t = (\theta_t, \phi_t)$ and $o_{\phi_t}$ is the linear output layer (with parameter $\phi_t$) on top of $h_{\theta_t}$. Then, only $o_{\phi_t}$ is supervised trained (with frozen $h_{\theta_t}$) using all the training dataset $\mathcal{D}_{1:t}$, including the labels, and the accuracy of resulting $f_{\Theta_t}$ becomes the proxy for the represenation quality.

**Class-/Data-/Domain-incremental learning.** We consider the three scenarios of continual learning as outlined in Van de Ven & Tolias (2019); Wang et al. (2023); Fini et al. (2022). We use $k$ and $j$ to denote arbitrary task numbers, where $k, j \in \{1, \ldots, T\}$ and $k \neq j$. The first category is the *class-incremental learning* (Class-IL), in which the $t$-th task's dataset consists of a unique set of classes for the input data, namely, $p(\mathcal{X}_k) \neq p(\mathcal{X}_j)$ and $\mathcal{Y}_k \cap \mathcal{Y}_j = \varnothing$. The second category is *domain-incremental learning* (Domain-IL), in which each dataset $\mathcal{D}_t$ has the same set of true labels but with different distribution on $\mathcal{X}_t$, denoted as $p(\mathcal{X}_k) \neq p(\mathcal{X}_j)$ but $\mathcal{Y}_k = \mathcal{Y}_j$. In other words, each dataset in Domain-IL contains input images sampled from a different domain, but the corresponding set of true labels is the same as for other tasks. Finally, we consider *data-incremental learning* (Data-IL), in which a set of input images $\mathcal{X}_t$ is sampled from a single distribution, $p(\mathcal{X}_k) = p(\mathcal{X}_j)$, but $\mathcal{Y}_k = \mathcal{Y}_j$. To implement the Data-IL scenario in our experiments, we shuffled the entire dataset (such as ImageNet-100) and divided it into $T$ disjoint datasets.

### 3.2 MOTIVATION

**Conventional loss form of continual learning** We first reconsider the loss form, illustrated in Figure 1(a), which is commonly employed by many state-of-the-art continual learning (CL) methods, except in cases where exemplar memory is utilized. Namely, when learning task $t$ with a model parameter $\Theta_t$, the conventional loss has the form as follows:

$$\mathcal{L}_{\mathrm{CL}}^t(\mathcal{D}_t, \Theta_t, \Theta_{t-1}) = \mathcal{L}_{\mathrm{single}}(\mathcal{D}_t, \Theta_t) + \lambda \cdot \mathcal{L}_{\mathrm{reg}}(\mathcal{D}_t, \Theta_t, \Theta_{t-1}), \tag{1}$$

in which $\Theta_{t-1}$ represents the frozen model trained up to task $t-1$ and $\lambda$ denotes a regularization parameter that controls appropriate trade-off between the two loss terms. The term $\mathcal{L}_{\mathrm{single}}$ represents the typical loss function utilized in single-task learning, such as cross-entropy, supervised contrastive loss (Khosla et al., 2020) or self-supervised learning loss (He et al., 2020). This term clearly promotes *plasticity* for the new task (task $t$) since it depends only on the current model $\Theta_t$ and $\mathcal{D}_t$. In addition, $\mathcal{L}_{\mathrm{reg}}$ represents a regularization term that is dedicated to increase *stability*. Namely, since $\Theta_{t-1}$ is considered to contain knowledge of the previous tasks' dataset (*i.e.*, $\mathcal{D}_{1:t-1}$) in the weight parameters, $\mathcal{L}_{\mathrm{reg}}$ is solely designed to maintain that knowledge to reduce forgetting while training $\Theta_t$ in various ways: such as direct weight level-regularization (Kirkpatrick et al., 2017), knowledge distillation (Li & Hoiem, 2017) from $\Theta_{t-1}$ to $\Theta_t$, and model expansion term (Yan et al., 2021). It's worth noting that recent works on CSSL (Hu et al., 2022; Fini et al., 2022; Madaan et al., 2022) also adopt a similar

---

[1] For concreteness, we explicitly work with image data in this paper, but we note that our method is general and not confined to image modality.

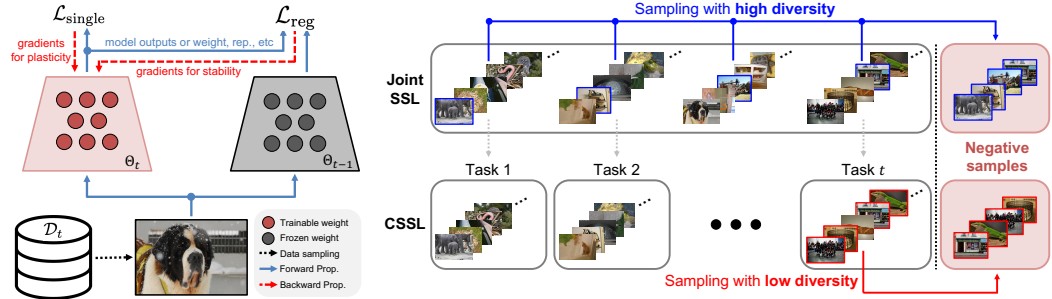

(a) Conventional loss form of CL  (b) Lack of diversity of negative samples in CSSL

Figure 1: Illustrations of motivation for our work. (a) Many of the CL algorithms combine a loss dedicated to learning a single current task and a regularizer that penalizes the deviation of the current model from the past. (b): In CSSL, negative samples are solely mined from the current task's data, resulting in significantly decreasing diversity of negative samples than the case of the joint SSL.

loss formulation as (1). In particular, CaSSLe (Fini et al., 2022), the current state-of-the-art method, introduces a novel and effective $\mathcal{L}_{\text{reg}}$ that easily integrates with various self-supervised learning methods, effectively mitigating catastrophic forgetting in learned representations during CSSL.

Nevertheless, we contend that the utilization of such a conventional loss form may impose limitations on the achievement of more successful CSSL. This stems from the fact that while $\mathcal{L}_{\text{reg}}$ contributes to the preservation of previously acquired representations and the reduction of forgetting , it may impede the acquisition of new representations from novel tasks. Furthermore, recent empirical findings, as highlighted in studies such as (Cha et al., 2023; Davari et al., 2022), suggest that prioritizing *stability* through regularization ($\mathcal{L}_{\text{reg}}$) may not necessarily result in learning improved representations even when higher accuracy is attained, thereby supporting this proposition.

**Low diversity of negative samples in CSSL**   In contrast to the prevailing research trend in CSSL, we assert that CSSL faces a critical challenge stemming from the *limited diversity of negative samples* when compared to joint self-supervised learning (Joint SSL). The importance of negative samples in self-supervised contrastive learning has been extensively acknowledged, with studies demonstrating the benefits of incorporating a substantial number of diverse negative samples (Oord et al., 2018; Wang & Isola, 2020; Chen et al., 2020a; Tao et al., 2022) and selecting hard negatives (Robinson et al., 2020; Kalantidis et al., 2020). However, the issue of negative samples in the CSSL has remained unexplored until now. In this regard, Figure 1(b) highlights a fundamental problem: while Joint SSL allows for diverse negative sample selection, CSSL tends to exhibit significantly reduced diversity. Given the importance of negative samples, we believe that this decrease in diversity can be a key reason why self-supervised contrastive learning algorithms often yield less effective representations in CSSL scenarios, especially in terms of *plasticity*.

### 3.3 AUGMENTING NEGATIVE REPRESENTATIONS FOR CONTRASTIVE LEARNING METHODS

Motivated by the two issues raised in the previous section, we propose a new form of loss functions tailored for CSSL. First, recalling the notation in Section 3.1, we note that only the encoder $h_{\theta_t}$ is trained during CSSL. Then, our Augmented Negatives (AugNeg) loss, devised for contrastive learning-based methods, for learning from task $t$ has the following form

$$\mathcal{L}^t_{\text{AugNeg}}(\mathcal{D}_t, \theta_t, \theta_{t-1}) = \mathcal{L}_1(\mathcal{D}_t, \theta_t, \theta_{t-1}) + \mathcal{L}_2(\mathcal{D}_t, \theta_t, \theta_{t-1}). \tag{2}$$

While we will specify the two loss functions shortly, here, notice the symmetric dependence of them on $(\mathcal{D}_t, \theta_t, \theta_{t-1})$, which is in contrast to (1). As we show below, such symmetry enables our loss to promote finding inherent trade-off between the plasticity and stability without explicit control of a hyperparameter.

**Model-based augmentation of negative representations.**   To address the issue depicted in Figure 1(b) and based on the format of Equation (2), we propose to make use of augmented negative representations obtained from the both current and previous models during training task $t$. For given input data $x_i$ and encoder $h_{\theta_t}$, we denote $z_{i,t} = \text{Proj}(h_{\theta_t}(x_i)) \in \mathcal{R}^{D_P}$ and $\tilde{z}_{i,t} = \text{Pred}(z_{i,t}) \in \mathcal{R}^{D_P}$ as the normalized embeddings for the encoder of task $t$, in which "Proj" and "Pred" are the

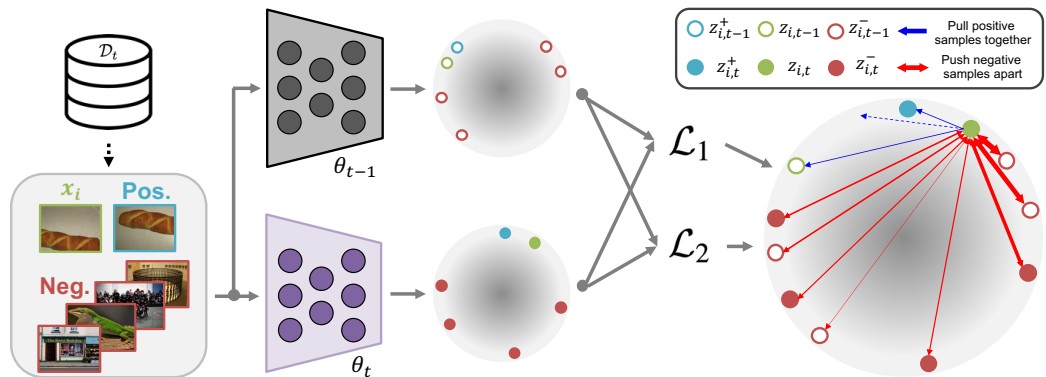

Figure 2: Graphical representation of learning with our proposed AugNeg loss. The blue dashed arrow represents the direction of the gradient update when training with the AugNeg loss, repelling from the negative sample embeddings of current/past models and attracting to the positive sample embeddings of the current/past models.

Projection (Chen et al., 2020a) and Prediction (Fini et al., 2022) layers, consisting of a multi-layer perceptron (MLP), respectively. Furthermore, we denote $z_{i,t}^+$ and $z_{i,t}^-$ as the embedding (obtained by $h_{\theta_t}$) of the *positive* and *negative* sample corresponding to the given $x_i$, respectively, and $\mathcal{N}_t(i)$ as the set of the negative embeddings for $x_i$. Then, both losses in (2) for the sample $x_i$ are defined as

$$\mathcal{L}_1(x_i, \theta_t, \theta_{t-1}) = -\log \frac{\exp(z_{i,t} \cdot z_{i,t}^+/\tau)}{\sum_{z_{i,t}^- \in \mathcal{N}_t(i)} \exp(z_{i,t} \cdot z_{i,t}^-/\tau) + \sum_{z_{i,t-1}^- \in \mathcal{N}_{t-1}(i)} \exp(z_{i,t} \cdot z_{i,t-1}^-/\tau)},$$

and

$$\mathcal{L}_2(x_i, \theta_t, \theta_{t-1}) = -\log \frac{\exp(\tilde{z}_{i,t} \cdot z_{i,t-1}/\tau)}{\sum_{z_{i,t}^- \in \mathcal{N}_t(i)} \exp(\tilde{z}_{i,t} \cdot z_{i,t}^-/\tau) + \sum_{z_{i,t-1}^- \in \mathcal{N}_{t-1}(i)} \exp(\tilde{z}_{i,t} \cdot z_{i,t-1}^-/\tau)}.$$

in which $\tau$ is a temperature parameter.

Note these two losses are quite similar in form to InfoNCE (Oord et al., 2018), but has a couple of key differences. Previous CL research, such as CaSSLe, $z_{i,t-1}$ was solely utilized for distilling previous knowledge (for better *stability*). In contrast to them, firstly, in $\mathcal{L}_1$, we use *additional* negative embeddings in $\mathcal{N}_{t-1}(i)$ which are obtained from the previous model $h_{\theta_{t-1}}$. This addition of negative embeddings in $\mathcal{L}_1$ mitigates the issue of decreasing diversity of negative samples pointed out in Section 3.2. Moreover, it compels the embedding of $x_i$ to be repelled not only from the negative embeddings of the current model but also from those of the previous model, hence, it fosters the acquisition of more distinctive representations. Secondly, in $\mathcal{L}_2$, which has the similar form of self-supervised distillation (Fang et al., 2021; Tian et al., 2019), we also use *additional* negative embeddings in $\mathcal{N}_t(i)$. Such modification has an impact of putting additional constraint in distillation so that the representations from the past model are maintained in a way not to contradict with the representations of current model. Also, we introduce the "Pred" layer, as in Fini et al. (2022), to not hurt the plasticity by doing indirect distillation.

Note that the denominators of the two loss functions are identical except for the "Pred" step in $\mathcal{L}_2$. Therefore, with the identical repelling negative embeddings, adding two losses will result in achieving natural trade-off between plasticity and stability for learning the representation $z_{i,t}$. This intuition is depicted in Figure 2. Namely, unlike in Figure 1(a), both losses symmetrically consider the embeddings from current and previous models, and as shown in the rightmost hypersphere, the representation of $z_{i,t}$ gets attracted to $z_{i,t}^+$ and $z_{i,t-1}$ with the constraint that it is far from both $z_{i,t}^-$ and $z_{i,t-1}^-$. Thus, the new representation will be distinctive from the previous model (*plasticity*) and carry over the old knowledge (*stability*) in a way not to hurt the current model.

It is noteworthy that the AugNeg loss can be easily integrated into contrastive learning-based SSL methods like MoCo and SimCLR. The implemented loss function for them is presented in the Supplementary Materials. Also, in the Supplementary Materials, we provide more discussions and analyses, such as the gradient analysis for the proposed loss $\mathcal{L}_{\text{AugNeg}}$, the role of augmented negatives.

### 3.4 EXTENSION TO THE NON-CONTRASTIVE LEARNING METHODS

For SSL algorithms that do not utilize negatives (e.g., Barlow, BYOL and VICReg), the core idea of AugNeg which utilizes model-based augmentation of negative representations cannot be directly applied. However, we can apply the similar principle by adding an additional regularization term. For instance, consider a batch of input images denoted as $x$, where distinct augmentations, denoted as $A$ and $B$, are applied, resulting in $x^A$ and $x^B$, respectively. Based on this, the loss function for the AugNeg loss s BYOL can be denoted as follows:

$$\mathcal{L}^t_{\text{AugNeg, BYOL}}(x^{\text{A}}, x^{\text{B}}, \theta_t, \theta_{t-1}) = \|g_{\theta_t}(h_{\theta_t}(x^{\text{A}})) - z_\xi(x^{\text{B}})\|^2_2 \tag{3}$$

$$+\|\text{Pred}(h_{\theta_t}(x^{\text{A}})) - h_{\theta_{t-1}}(x^{\text{A}})\|^2_2 \tag{4}$$

$$-\|\text{Pred}(h_{\theta_t}(x^{\text{A}})) - h_{\theta_{t-1}}(x^{\text{B}})\|^2_2, \tag{5}$$

where $q_{\theta_t}$ represents an MLP layer in BYOL, and $z_\xi$ corresponds to the target network obtained through momentum updates using $h_\theta$. Equation (3) and Equation (4) denotes the original loss function of BYOL and CaSSLe's distillation, respectively. Furthermore, Equation (5) is a newly devised regularization to enhance *plasticity* by using negatives from the $t-1$ model. Unlike Equation (4), where the outputs of models $t$ and $t-1$ task remain consistent when the same augmentation is applied, Equation (5) ensures that when different augmentations are applied, model $t$ produces different outputs from model $t-1$ (i.e., considering the output of model $t-1$ as negative samples). The justification for Equation (5) can be substantiated by the approach utilized in contrastive distillation of CaSSLe, where all samples with distinct augmentations (e.g., $2N$ samples) consider all samples except those with the same augmentation (e.g., $2N-1$ samples) as negative samples (See the Supplementary Materials for more details). Note that the similar form of Equation 5 can be utilized for the case of AugNeg(VICReg) also. We believe that the above idea can similarly be applied to various SSL algorithms that do not explicitly involve negative samples through the addition of extra regularization. The implementation details for VICReg and Barlow are introduced in the Supplementary Materials.

## 4 EXPERIMENTS

### 4.1 EXPERIMENTAL DETAILS

**Baselines**    To evaluate the AugNeg loss, we used CaSSLe (Fini et al., 2022) as our primary baseline, which has shown state-of-the-art performance in CSSL. We selected four SSL methods, SimCLR (Chen et al., 2020a), MoCo v2 Plus (MoCo) (Chen et al., 2020b), BarlowTwins (Barlow) (Zbontar et al., 2021), BYOL (Grill et al., 2020), and VICReg (Bardes et al., 2022), which achieve superior performance with the combination with CaSSLe in various CSSL scenarios.

**Implementation details**    We conducted experiments on three datasets: CIFAR-100 (Krizhevsky et al., 2009), ImageNet-100 (Deng et al., 2009), and DomainNet (Peng et al., 2019), following the methodology outlined in Fini et al. (2022). For CIFAR-100 and ImageNet-100, we performed Class- and Data-incremental learning (Class- and Data-IL) with 5 and 10 tasks (denoted as 5T and 10T), respectively. For Domain-incremental learning (Domain-IL), we utilized DomainNet, consisting of six disjoint datasets from different six source domains. The ResNet-18 (He et al., 2016) model implemented in PyTorch was used for all experiments. AugNeg was implemented based on the code provided by CaSSLe (Fini et al., 2022). We conducted all experiments of both CaSSLe and AugNeg in the unified environment for fair comparison. Each experiment was conducted three times with different seeds, and the reported values represent the mean, with standard deviation indicated in parentheses (). Further experimental details are available in the Supplementary Materials.

**Evaluation metrics**    To gauge the quality of representations learned in CSSL, we conduct linear evaluation by training only the output layer on the given dataset while maintaining the encoder $h_{\theta_t}$ as a fixed component, following (Fini et al., 2022; Cha et al., 2023). The average accuracy after learning the task $t$ is denoted as $A_t = \frac{1}{t}\sum_{i=1}^{T}a_{i,t}$, where $a_{i,j}$ stands for the linear evaluation top-1 accuracy of the encoder on the dataset of task $i$ after the end of learning task $j$. Furthermore, we employ measures of stability ($S$) and plasticity ($P$), and a comprehensive explanation of these terms is provided in the Supplementary Materials.

## 4.2 EXPERIMENTAL EVALUATION OF AUGNEG WITH SSL ALGORITHMS

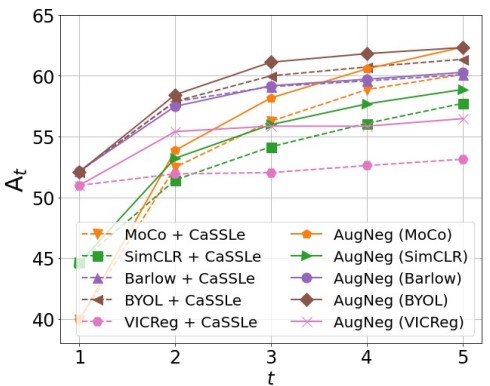

| $A_5$ | MoCo | SimCLR | Barlow | BYOL | VICReg |
|---|---|---|---|---|---|
| Joint | 66.90 (0.11) | 63.78 (0.22) | 68.99 (0.21) | 69.36 (0.28) | 68.01 (0.36) |
| FT | 51.95 (0.26) | 48.97 (0.74) | 55.81 (0.57) | 52.43 (0.62) | 52.43 (0.62) |
| EWC* | - | 53.60 | 56.70 | - | 56.40 |
| DER* | - | 50.70 | 55.30 | - | 54.80 |
| LUMP* | - | 52.30 | 57.80 | - | 56.40 |
| Less-Forget* | - | 52.50 | 56.40 | - | 58.60 |
| CaSSLe | 60.11 (0.30) | 57.73 (1.07) | 60.10 (0.38) | 61.36 (1.38) | 53.13 (0.04) |
| **AugNeg** | **62.36 (0.29)** | **58.87 (0.16)** | **60.28 (0.36)** | **62.32 (0.98)** | **56.47 (0.94)** |

Figure 3: Applying AugNeg to SSL methods.   Table 1: Experimental results on CSSL baselines.

To evaluate the effectiveness of integrating AugNeg with different SSL methods, we conducted experiments by incorporating the AugNeg into MoCo, SimCLR, Barlow, BTOL, and VICReg, as depicted in Figure 3. These experiments were carried out within the Class-IL (5T) scenario on CIFAR-100. In this figure, +CaSSLe indicates the outcomes of applying CaSSLe to SSL algorithm, and AugNeg denotes the results of implementing the AugNeg loss alongside SSL algorithm. Note that the training cost of CaSSLe and AugNeg is nearly identical, attributable to the inherent simplicity of AugNeg. This figure reveals several key findings. Firstly, when both CaSSLe and AugNeg are combined with each SSL method, they progressively enhance the quality of representations as each task $t$ is learned. Secondly, AugNeg demonstrates more effective integration with MoCo, SimCLR, BYOL, and VICReg, surpassing the results of CaSSLe.

Table 1 presents the numerical results for $A_5$ in the same scenario. In this table, Joint corresponds to the experimental results in the Joint SSL scenario and FT represents the results achieved through fine-tuning with the SSL method alone. The * symbol indicates results from the CaSSLe paper, while the others are reproduced outcomes. From this table, AugNet consistently outperforms CaSSLe, demonstrating a maximum improvement of approximately 2-3%. Especially, when considering the performance arising from the combination of CaSSLe with MoCo and BYOL, demonstrating superior performance in CSSL and already approaching that of the Joint, we assert that the additional performance enhancement achieved by AugNeg is a notable outcome.

Finally, Table 2 presents stability ($S$) and plasticity ($P$) metrics. For MoCo, SimCLR, BYOL, and VICReg, the results illustrate AugNeg's superiority, attributed to a significant boost in *plasticity* while

Table 2: Experimental results on stability and plasticity.

| $S(\downarrow)$ / $P(\uparrow)$ | MoCo | SimCLR | BYOL | VICReg |
|---|---|---|---|---|
| CaSSLe | 0.11 / 9.99 | 0.33 / 4.08 | 0.09 / 11.94 | 0.39 / 6.78 |
| AugNeg | 0.06 / 11.8 | 0.48 / 5.36 | 0.20 / 13.84 | 0.65 / 12.59 |

minimizing increase of *stability* compared to CaSSLe. This underscores the crucial role of employing the proposed model-based augmentation of negatives in CSSL. Based on the results of these previous experiments, we will more focus to apply and evaluate AugNeg(MoCo) and AugNeg(BYOL), which achieves superior performance, in further experiments on various CSSL scenarios.

## 4.3 EXPERIMENTAL EVALUATION IN DIVERSE CSSL SCENARIOS

**Class-IL**  We conducted experiments on various datasets and scenarios, and all the experimental results are presented in Table 3. The figure presents Class-IL experiments conducted with CIFAR-100 and ImageNet-100 datasets. Firstly, regarding the CaSSLe results, significant performance variations are observed depending on the dataset and the total number of tasks. For instance, in the CIFAR-100 experiments, BYOL + CaSSLe consistently outperforms other CaSSLe variants. However, in the ImageNet-100 experiments, Barlow + CaSSLe exhibits superior performance in 5T, whereas BYOL + CaSSLe shows the lowest performance. On the other hand, AugNeg combined with MoCo and BYOL consistently achieves improved performance compared to their combination with CaSSLe. For example, in the ImageNet-100 experiments, AugNeg(MoCo) and AugNeg(BYOL) achieves a substantial gain of approximately 2-6% when compared to their combination with CaSSLe, establishing themselves as the new state-of-the-art method.

Table 3: Experimental results for three scenarios. All results are averaged over three seeds, and standard deviations are indicated in parentheses (). The red annotation represents the highest performance, while the blue annotation indicates the second-highest performance, in each scenario.

| $A_T$ | | Class-IL | | | Data-IL | | Domain-IL |
|---|---|---|---|---|---|---|---|
| | | CIFAR-100 | ImageNet-100 | | ImageNet-100 | | DomainNet |
| | | 10T | 5T | 10T | 5T | 10T | 6T |
| MoCo | Joint | 66.90 (0.11) | 76.67 (0.56) | | | | 48.2 (0.30) |
| | FT | 34.11 (0.90) | 57.88 (0.35) | 48.27 (0.88) | 65.51 (0.72) | 60.50 (0.92) | 36.48 (1.01) |
| | +CaSSLe | 53.58 (0.41) | 63.49 (0.52) | 52.71 (0.47) | 66.88 (0.32) | 59.72 (0.61) | 38.04 (0.24) |
| SimCLR | Joint | 63.78 (0.22) | 71.91 (0.57) | | | | 48.5 (0.21) |
| | FT | 39.48 (1.00) | 56.11 (0.57) | 46.66 (0.59) | 62.88 (0.30) | 56.47 (0.11) | 39.46 (0.20) |
| | +CaSSLe | 53.02 (0.47) | 62.53 (0.11) | 54.55 (0.12) | 66.05 (0.95) | 61.68 (0.38) | 46.58 (0.08) |
| Barlow | Joint | 69.36 (0.21) | 75.89 (0.22) | | | | 49.5 (0.32) |
| | FT | 49.46 (0.67) | 60.27 (0.27) | 51.83 (0.46) | 66.47 (0.24) | 59.48 (1.33) | 41.87 (0.17) |
| | +CaSSLe | 54.46 (0.24) | 64.98 (0.79) | 56.27 (0.63) | 69.24 (0.36) | 63.12 (0.28) | 48.49 (0.04) |
| BYOL | Joint | 68.99 (0.28) | 75.52 (0.17) | | | | 53.8 (0.24) |
| | FT | 46.13 (0.88) | 60.77 (0.62) | 51.04 (0.52) | 69.76 (0.45) | 61.39 (0.44) | 47.29 (0.08) |
| | +CaSSLe | 57.36 (0.86) | 62.31 (0.09) | 57.47 (0.75) | 66.22 (0.13) | 63.33 (0.19) | 51.02 (0.08) |
| AugNeg (MoCo) | | 56.62 (0.31) | 67.85 (0.44) | 60.75 (0.39) | 69.98 (0.30) | 67.83 (0.45) | 43.86 (0.17) |
| AugNeg (BYOL) | | 58.44 (0.40) | 64.23 (0.37) | 60.11 (0.91) | 64.83 (0.13) | 61.90 (0.33) | 51.96 (0.08) |

**Data/Domain-IL**  In the context of Data-IL, baseline methods exhibits distinct trends from those observed in Class-IL. Specifically, Barlow + CaSSLe outperforms other variants. Moreover, as previously reported in the CaSSLe paper, fine-tuning with BYOL produces robust results. However, AugNeg (MoCo) shows a noteworthy improvement of approximately 3-8% when compared to MoCo + CaSSLe, achieving the state-of-the-art performance in Data-IL for both 5T and 10T scenarios. Note that combining BYOL with CaSSLe and AugNeg in Data-IL, especially in the 5T scenario, yields suboptimal results. This can be attributed to the unique characteristics of Data-IL and we discuss this result more in the Supplementary Materials.

In Domain-IL, we presented the average top-1 accuracy achieved by training a linear classifier independently on each domain, using the feature extractor that was kept frozen (domain-aware). BYOL + CaSSLe exhibits the most promising results among the existing baselines. In this scenario as well, the application of AugNeg consistently improves the performance of MoCo and BYOL compared to the baseline. Notably, AugNeg(BYOL) outperforms the previous state-of-the-art BYOL + CaSSLe, more approaching the performance of Joint.

**Challenges of reproducing CaSSLe's results**  While all experiments were conducted using CaSSLe's official code, we obtained results approximately 4-5% lower than those reported in the original paper, particularly for datasets such as ImageNet-100 and DomainNet. Notably, similar issues were documented on the issues page of CaSSLe's official GitHub repository. However, our experiments, which compared CaSSLe with our proposed AugNeg, were executed within an identical environment. This allowed us to demonstrate the superior performance of AugNeg without the confounding factors associated with fair comparison.

## 4.4  EXPERIMENTAL ANALYSIS FOR AUGNEG

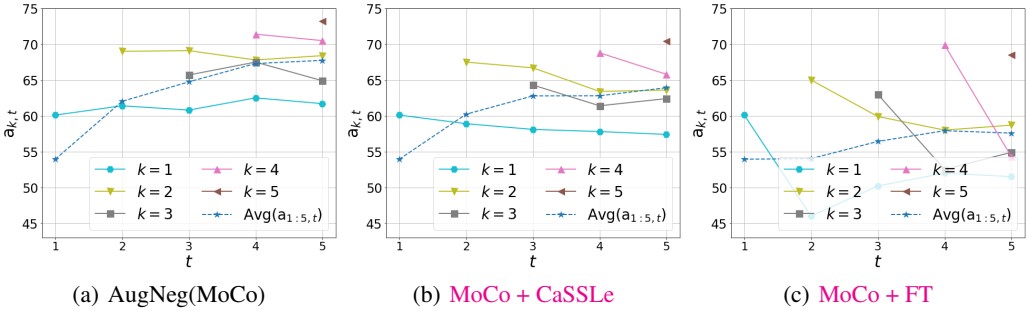

(a) AugNeg(MoCo)  (b) MoCo + CaSSLe  (c) MoCo + FT

Figure 4: The graph illustrates the values of $a_{k,t}$ for each algorithm in the Class-IL (5T) scenario using the ImageNet-100 dataset. The measured stability ($S \downarrow$) and plasticity ($P \uparrow$) for each method are as follows: (a) $(S, P) = (1.23, 3.47)$, (b) $(S, P) = (2.80, 2.52)$, (c) $(S, P) = (3.13, 2.38)$.

**Analysis on performance changes of each task**  Figure 4 presents the experimental results for Class-IL (5T) using the ImageNet-100 dataset, showcasing graphs of $a_{k,t}$ and $Avg(a_{1:5,t})$. To begin, in Figure

4(a) showing the result of AugNeg(MoCo), we observe a general upward trend in $a_{k,t}$ across all tasks. Remarkably, the performance for the initial task ($a_{k=1,t}$) remains relatively stable and even exhibits slight improvement as subsequent tasks are learned. Conversely, the results depicted in Figures 4(b) and 4(c) for MoCo + CaSSLe and MoCo + FT indicate that, while their Avg($a_{1:5,t}$) gradually increases, certain task performances experience gradual declines (e.g., $a_{k=2,t}$ of MoCo + CaSSLe and most $k$ of MoCo + FT), showing suffering from catastrophic forgetting than AugNeg(MoCo). Furthermore, the numerical assessments of plasticity and stability for each algorithm, as detailed in the caption of Figure 4, confirm that AugNeg(MoCo) achieves its performance improvement through superior *plasticity* and *stability* compared to other baselines. The equivalent analysis for other baselines can be found in the Supplementary Materials.

Table 4: Experimental results on downstream tasks. The red annotation represents the highest performance, while the blue annotation indicates the second-highest performance.

| Case | Data | MoCo + CaSSLe | SimCLR + CaSSLe | Barlow + CaSSLe | BYOL + CaSSLe | AugNeg (MoCo) | AugNeg (BYOL) |
|---|---|---|---|---|---|---|---|
| Semi-supervised | 10% | 56.48 | 55.16 | 55.10 | 54.22 | 61.74 | 57.36 |
| | 1% | 39.14 | 40.86 | 41.90 | 36.86 | 46.48 | 40.82 |
| Downstream | Three datasets | 58.61 | 56.73 | 58.13 | 61.35 | 62.04 | 62.53 |
| | Clipart | 28.32 | 34.68 | 37.42 | 38.98 | 38.86 | 41.57 |

**Evaluation for semi-supervised learning scenario and downstream tasks** We conducted experiments in a more practical semi-supervised setting. Specifically, we considered a scenario where a linear classifier is only trained using only 1% or 10% of the entire supervised ImageNet-100 dataset in the CSSL of Class-IL (5T). The experimental results are presented in the upper rows of Table 4. Notably, when compared to CaSSLe, applying AugNeg to both MoCo and BYOL yields approximately 3-6% performance improvements in both settings. As a result, AugNeg(MoCo) achieves a new state-of-the-art performance. The lower rows of Table 4 present the results of linear evaluation for downstream tasks conducted on the same encoders. Firstly, for the three datasets, we reported the average accuracy of linear evaluation results on STL-10 (Coates et al., 2011), CIFAR-10, and CIFAR-100 (Krizhevsky et al., 2009) datasets. Secondly, we set the downstream task to Clipart within the DomainNet (Peng et al., 2019) dataset and reported the results of linear evaluation using it. From these experimental outcomes, we once again demonstrate that both AugNeg(MoCo) and AugNeg(BYOL) achieve performance improvements compared to the CaSSLe results.

Moreover, we present supplementary experimental results in the Supplementary Materials, encompassing more detailed results and exploring variations in queue size, mini-batch size, and others.

**Ablation study** We present the results of the ablation study conducted on CIFAR-100 in the Class-IL (5T) setting, as illustrated in Table 5. The first row represents the performance of AugNeg(MoCo) when all additional negative samples ($\mathcal{N}_t$ and $\mathcal{N}_{t-1}$) are included in both $\mathcal{L}_1$ and $\mathcal{L}_2$. Case 1 to 3 demonstrate the results when one of $\mathcal{N}_t$ and $\mathcal{N}_{t-1}$ is excluded or when both are omitted.

Our experimental results confirm that the absence of additional negative samples leads to a gradual reduction in performance, indicating that AugNeg(MoCo) acquires superior representations by leveraging these additional negative samples from each model, $h_{\theta_t}$ and $h_{\theta_{t-1}}$. Moreover, the results from Case 4, which uses a queue size twice as large for MoCo

Table 5: Ablation study

| | $\mathcal{N}_t$ | $\mathcal{N}_{t-1}$ | + CaSSLe | |queue| $\times 2$ | $A_5$ |
|---|---|---|---|---|---|
| AugNeg | ✓ | ✓ | ✗ | ✗ | 62.36 |
| Case 1 | ✗ | ✓ | ✗ | ✗ | 61.26 |
| Case 2 | ✓ | ✗ | ✗ | ✗ | 60.92 |
| Case 3 | ✗ | ✗ | ✗ | ✗ | 59.97 |
| Case 4 | ✗ | ✗ | ✓ | ✓ | 60.09 |

+ CaSSLe, illustrate that integrating the extra negative samples from each model is distinct from merely enlarging the queue size. Additionally, note that the ablation study for AugNeg(BYOL) can be conducted by comparing the results between BYOL + CaSSLe and AugNeg(BYOL) in Table 3.

## 5 CONCLUDING REMARKS

We introduce the Augmented Negatives (AugNeg) loss, a simple but novel approach to augmenting negative representations in continual self-supervised learning (CSSL). Initially, we highlight the limitations of the traditional CSSL loss formulation and the lack of diversity in negative samples during CSSL. To address these challenges, we propose considering additional negatives from both previous and current models for InfoNCE-based contrastive learning. Furthermore, we show that the idea of AugNeg can be extended to non-contrastive methods by adding extra regularization. Through extensive experiments, we observe that not only our AugNeg can be applied to various self-supervised learning methods but also achieves state-of-the-art performance with superior stability and plasticity.

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

# A    SUPPLEMENTARY MATERIALS FOR SECTION 3

## A.1    ANALYSIS OF THE GRADIENTS

Here, we give the gradient analysis of our AugNeg loss for contrastive learning methods. For simplicity, we assume that "Pred" is an identity map instead of a MLP, hence $\tilde{z}_{i,t} = z_{i,t}$. Now, as given in the Supplementary Material, we can show the gradient of $\mathcal{L}_{\text{AugNeg}}$ with respect to $z_{i,t}$ becomes

$$\frac{\partial(\frac{1}{2}\mathcal{L}_{\text{AugNeg}}(x_i))}{\partial z_{i,t}} = -\underbrace{\left(\frac{z_{i,t}^+ + z_{i,t-1}}{2}\right)}_{(a)} + \underbrace{\left\{\sum_{z_{i,t}^- \in \mathcal{N}_t(i)} z_{i,t}^- \cdot S_{i,t}(z_{i,t}^-) + \sum_{z_{i,t-1}^- \in \mathcal{N}_{t-1}(i)} z_{i,t-1}^- \cdot S_{i,t}(z_{i,t-1}^-)\right\}}_{(b)},$$

in which $S_{i,t}(u) = \exp(z_{i,t} \cdot u)/\left[\sum_{z_{i,t}^- \in \mathcal{N}_t(i)} \exp(z_{i,t} \cdot z_{i,t}^-/\tau) + \sum_{z_{i,t-1}^- \in \mathcal{N}_{t-1}(i)} \exp(z_{i,t} \cdot z_{i,t-1}^-/\tau)\right]$, and $\sum_{z_{i,t}^- \in \mathcal{N}_t(i)} S_{i,t}(z_{i,t}^-) + \sum_{z_{i,t-1}^- \in \mathcal{N}_{t-1}(i)} S_{i,t}(z_{i,t-1}^-) = 1$. Similarly as in the Unified Gradient of the InfoNCE loss (Tao et al., 2022), we can make the following interpretations. Namely, the negative gradient step can be decomposed into two parts, part (a) and the negative of part (b) above.

Part (a) is the average of the embedding of $h_{\theta_t}$ for the positive sample and the embedding of $h_{\theta_{t-1}}$ for the input sample (*i.e.*, $x_i$). Hence, this direction encourages the model to learn new representations while taking the *stability* from the previous model into account. On the other hand, negative of part (b) is the repelling direction from the center of mass point among the negative sample embeddings in $\mathcal{N}_t(i) \cup \mathcal{N}_{t-1}(i)$, in which each element $u \in \mathcal{N}_t(i) \cup \mathcal{N}_{t-1}(i)$ has the probability mass $S_{i,t}(u)$. Thus, this direction promotes the new representations to be more discriminative from the current and previous models' negative sample embeddings, leading to improved *plasticity*. Our gradient analysis allows us to better understand the graphical representation in Figure 2.

## A.2    DISCUSSION: THE ROLE OF AUGMENTED NEGATIVE REPRESENTATIONS

Incorporating augmented negative representations from the previous model (referred to as the $t-1$ model) aligns with SSL algorithms, which leverage positive representations from a momentum encoder, such as BYOL and MoCo. This momentum encoder progressively enhances their representations by assimilating updates from the training encoder, thus contributing to the successful *positive* representations for SSL. In contrast, in CSSL, the current task model (typically initialized from the $t-1$ model) is trained by leveraging information from the previous model with CSSL algorithm, such as CaSSLe (Fini et al., 2022). It's important to note that, while the previous model may have achieved adequate uniformity and alignment for the previous task dataset (Wang & Isola, 2020), it may not exhibit the same characteristics when applied to a new task, as shown in Figure 2. Therefore, in scenarios involving knowledge distillation (similar to CaSSLe), merely distilling the representations of samples from the current task dataset, such as representations of the same image with the same augmentation (*e.g.*, the numerator of $L_2$ in Section 3.3 and Equation (4)) may have a limitation to achieve a superior representation.

In this regard, using augmented negative representations from the previous model can be considered as contributing to the informative additional *negative* representations for CSSL. By additionally incorporating representations from both the models (*e.g.*, the denominator of $L_1$ and $L_2$ in Section 3.3) or the previous model (*e.g.*, Equation (5)) as supplementary negative representations, the newly trained model can learn superior and novel representations (better *plasticity*), taking into account both uniformity and alignment of not only the current but also previous model.

## A.3    DETAILED LOSS FUNCTION OF AUGNEG WITH SSL METHODS

In this section, we will delve into the detailed loss function of AugNeg in conjunction with SSL methods, such as AugNeg with MoCo (He et al., 2020), SimCLR (Chen et al., 2020a), and VI-CReg (Bardes et al., 2022). It's important to note that the details of AugNeg(BYOL) are already provided in Section 3.4. We will adhere to the notations introduced in Section 3.4 and the original formulation of the loss function for each SSL method.

### A.3.1 AUGNEG(MOCO)

AugNeg(MoCo) can be represented by the equation $\mathcal{L}_{\text{AugNeg}}^t$ proposed in Section 3.3. Specifically, $\mathcal{L}1^{\text{MoCo}}$ and $\mathcal{L}_2^{\text{MoCo}}$ are as follows:

$$\mathcal{L}_1^{\text{MoCo}}(x_i, \theta_t, \theta_{t-1}) = -\log \frac{\exp(z_{i,t} \cdot z_{i,t}^+/\tau)}{\sum_{z_{i,t}^- \in \mathcal{N}_t(i)} \exp(z_{i,t} \cdot z_{i,t}^-/\tau) + \sum_{z_{i,t-1}^- \in \mathcal{N}_{t-1}(i)} \exp(z_{i,t} \cdot z_{i,t-1}^-/\tau)} \tag{6}$$

$$\mathcal{L}_2^{\text{MoCo}}(x_i, \theta_t, \theta_{t-1}) = -\log \frac{\exp(\tilde{z}_{i,t} \cdot z_{i,t-1}/\tau)}{\sum_{z_{i,t}^- \in \mathcal{N}_t(i)} \exp(\tilde{z}_{i,t} \cdot z_{i,t}^-/\tau) + \sum_{z_{i,t-1}^- \in \mathcal{N}_{t-1}(i)} \exp(\tilde{z}_{i,t} \cdot z_{i,t-1}^-/\tau)} \tag{7}$$

Note that in the MoCo-based implementation of AugNeg, both $\mathcal{N}_t$ and $\mathcal{N}_{t-1}$ represent queues that store positives from the previous iterations of $t$ and $t-1$ models, respectively. The entire set of saved samples in each queue is utilized as negatives for the current iteration. In our experiments, we maintained a queue size of 65,536 consistent with the original MoCo (He et al., 2020) and CaSSLe (Fini et al., 2022).

### A.3.2 AUGNEG(SIMCLR)

AugNeg(SimCLR) is also derived from the equation $\mathcal{L}_{\text{AugNeg}}^t$. However, the key distinction with AugNeg(MoCo) lies in the fact that negatives from $t$ and $t-1$ models are sourced from the current mini-batch. When representing the $i$-th input image as $x_i$, applying distinct augmentations $A$ and $B$ to N input images results in an augmented batch $x' = [x_1^A, ..., x_N^A, x_{N+1}^B, ...x_{2N}^B]$ with $2N$ elements (where $i = 1, ..., N$ and $x_i = x_{i+N}$). Based on this, the SimCLR implementation of $\mathcal{L}_1^{\text{SimCLR}}$ and $\mathcal{L}_2^{\text{SimCLR}}$ in AugNeg is as follows:

$$\mathcal{L}_1^{\text{SimCLR}}(x_i, \theta_t, \theta_{t-1}) = -\log \frac{\exp(S(z_{i,t}, z_{i+N,t})/\tau)}{\sum_{k=1}^{2N} \mathbb{1}_{[k \neq i]} \exp(S(z_{i,t}, z_{k,t})/\tau) + \sum_{k=1}^{2N} \mathbb{1}_{[k \neq i]} \exp(S(z_{i,t}, z_{k,t-1})/\tau)} \tag{8}$$

$$\mathcal{L}_2^{\text{SimCLR}}(x_i, \theta_t, \theta_{t-1}) = -\log \frac{\exp(S(\tilde{z}_{i,t}, z_{i,t-1})/\tau)}{\sum_{k=1}^{2N} \mathbb{1}_{[k \neq i]} \exp(S(\tilde{z}_{i,t}, z_{k,t})/\tau) + \sum_{k=1}^{2N} \mathbb{1}_{[k \neq i]} \exp(S(\tilde{z}_{i,t}, z_{k,t-1})/\tau)}, \tag{9}$$

where $S(A, B) = (A)^\top(B)/(||(A)^\top|| \cdot ||(B)||)$ and $\mathbb{1}_{[k \neq i]}$ is indicator function. It's important to note that in the case of Supervised Contrastive Learning (SupCon) (Khosla et al., 2020), we can straightforwardly extend the AugNeg(SimCLR) to incorporate supervised labels, as proposed in SupCon.

### A.3.3 AUGNEG(VICREG)

To implement AugNeg(VICReg), we merely incorporate an additional regularization proposed in Section 3.4, similar to AugNeg(BYOL). Adhering to the original loss function proposed in Bardes et al. (2022) and the notations introduced in Section 3.4, AugNeg(VICReg) can be expressed as:

$$\mathcal{L}_{\text{AugNeg, VICReg}}^t(x^A, x^B, \theta_t, \theta_{t-1}) = \lambda s(z_t^A, z_t^B) + \mu[v(z_t^A) + v(z_t^B)] + \nu[c(z_t^A) + c(z_t^B)] \tag{10}$$
$$+ \lambda_{\text{CaSSLe}}[s(\tilde{z}_t^A, z_{t-1}^A) * 0.5 + s(\tilde{z}_t^B, z_{t-1}^B) * 0.5] \tag{11}$$
$$+ \lambda_{\text{AugNeg}}[s(\tilde{z}_t^A, z_{t-1}^B) * 0.5 + s(\tilde{z}_t^B, z_{t-1}^A) * 0.5], \tag{12}$$

where Equation (10) is the original VICReg loss function and $s(\cdot, \cdot)$ denotes the mean-squared euclidean distance between each pair of vectors. $\lambda_{\text{CaSSLe}}$ and $\lambda_{\text{AugNeg}}$ are hyperparameters. Equa-

tion (11) represents CaSSLe's distillation, while Equation (12) serves as an additional regularization for AugNeg in non-contrastive learning, sharing a similar form with Equation (5) used in AugNeg(BYOL).

For all algorithms, we only modified the loss function to take the form described above, while keeping the remaining elements consistent with the original training procedures of each SSL algorithm.

### A.4 MOTIVATION OF AUGNEG FOR NON-CONTRASTIVE LEARNING

Comparing $\mathcal{L}_2^{\text{SimCLR}}$ to Equation (4) in Section 3.4, we can consider that the numerator part $(\exp(\hat{S}_{i,i}/\tau))$ plays a role similar to Equation (4). However, since BYOL and VICReg do not fully consider negative samples for self-supervised learning, terms like the denominator of $\mathcal{L}_2^{\text{SimCLR}}$ do not exist.

The motivation behind the regularization defined in Equation (5) and Equation (12) for considering negatives from the t-1 model stems from one of the terms in the denominator of $\mathcal{L}_2^{\text{SimCLR}}$, namely $\sum_{k=1}^{2N} \mathbb{1}_{[k \neq i]}\exp(S(\tilde{z}_{i,t}, z_{k,t-1})/\tau)$. It considers all $2N - 1$ samples as negatives for the given augmented $x_i$ with the condition that $k \neq i$ excluding only one sample. For instance, among the $2N - 1$ negatives for $x_1^A$, an image with the different augmentation applied to the same source input $x$ (e.g., $x_{1+N}^B$) is included as the negatives. As a result, the distance between the output features of the $t$ model for $x_1^A$ and the $t - 1$ model for $x_{1+N}^B$ increases during training task $t$.

Both Equation (5) and Equation (12) are BYOL and VICReg implementations for considering the negatives mentioned above, respectively. More specifically, for images with different augmentations applied to the same source image $x$, denoted as $x^A$ and $x^B$, this additional regularization is applied to increase the squared mean square error between the output features of each $t$ and $t - 1$ model to ensure that they differ. With these explanations in mind, we would like to underscore that the additional regularization introduced to incorporate negatives from the $t - 1$ model for CSSL using non-contrastive learning is inspired by the AugNeg Loss presented in Section 3.3.

## B SUPPLEMENTARY MATERIALS FOR SECTION 4

### B.1 SUBOPTIMAL PERFORMANCE OF CASSLE AND AUGNEG IN DATA-IL

As shown in Table 3, combining BYOL with CaSSLe and AugNeg in Data-IL led to suboptimal performance, particularly in the 5T scenario. This can be attributed to the unique characteristics of Data-IL, briefly discussed in the CaSSLe paper. Data-IL involves shuffling and evenly distributing the ImageNet-100 dataset among tasks, resulting in minimal distribution disparities between them. During BYOL training (using Equation (3)), the target encoder ($\xi$) retains some information from the current task which is conceptually similar to the previous task, due to momentum updates from the training encoder ($h_{\theta_t}$). As a result, fine-tuning solely with BYOL can produce a robust Data-IL outcome due to the substantial similarity in distribution between tasks. However, the incorporation of CaSSLe (Equation (4)) and AugNeg (Equation (4)) may conflict with Equation (3) in Data-IL.

On the contrary, when applied to the Domain-IL scenario using the DomainNet dataset where distinct variations in input distribution are evident for each task, AugNeg(BYOL) effectively demonstrates the feasibility of augmenting negative representations as outlined in Table 3. This reinforces our conviction that the suboptimal results observed in Data-IL are solely attributable to the unique and artificial circumstances inherent to Data-IL.

### B.2 MEASURES FOR STABILITY AND PLASTICITY

To evaluate each CSSL algorithm in terms of stability and plasticity, we use measures for them, following Fini et al. (2022); Cha et al. (2021b), as shown in below:

- Stability: $S = \frac{1}{T-1} \sum_{i=1}^{T-1} \max_{t \in \{1,...,T\}}(a_{i,t} - a_{i,T})$
- Plasticity: $P = \frac{1}{T-1} \sum_{j=1}^{T-1} \frac{1}{T-j} \sum_{i=j+1}^{T}(a_{i,j} - FT_i)$

Here, $FT_i$ signifies the linear evaluation accuracy (on the validation dataset of task $i$) of the model trained through finetuning with an base SSL algorithm until task $i$.

## B.3 ADDITIONAL EXPERIMENTAL ANALYSIS FOR AUGNEG (BYOL)

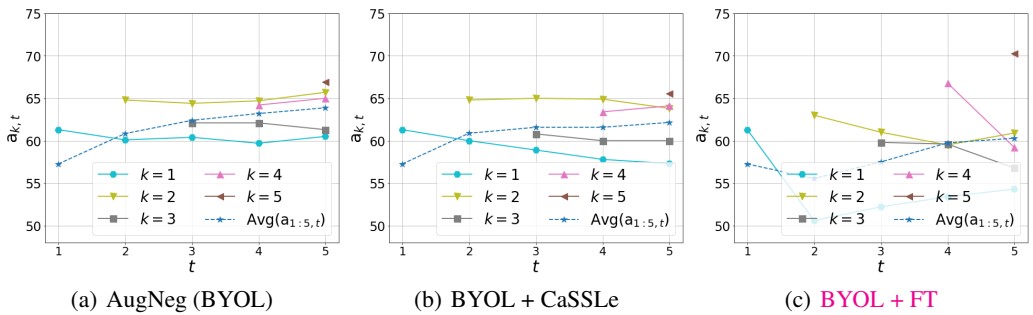

(a) AugNeg (BYOL)  (b) BYOL + CaSSLe  (c) BYOL + FT

Figure 5: The graph illustrates the values of $a_{k,t}$ for each algorithm in the Class-IL (5T) scenario. The measured stability and plasticity for each method are as follows: (a) $(S, P) = (0.4, -0.07)$, (b) $(S, P) = (1.5, -0.47)$, (c) $(S, P) = (4.9, -1.6)$.

Figure 5 depicts the results of the experimental analysis conducted on AugNeg (BYOL), BYOL + CaSSLe, and BYOL + FT in Class-IL (5T) experiments using the ImageNet-100 dataset. The trends observed here align with those presented in the manuscript. Initially, there is a gradual increase in their $\text{Avg}(a_{1:5,t})$. However, it is noteworthy that $a_{k=1,t}$, $a_{k=3,t}$ for BYOL + CaSSLe decrease across tasks. Similarly, BYOL + FT experiences a decline in most $k$. In contrast, the application of the proposed AugNeg not only maintains $a_{k=1,t}$ but also results in a gradual increase in $a_{k=2,t}$ and $a_{k=4,t}$, suggesting that AugNeg surpasses BYOL + CaSSLe in terms of plasticity and stability. The plasticity ($P \uparrow$) and stability ($S \downarrow$) measurements mentioned in the caption of Figure 5 further substantiate these experimental findings.

## B.4 ADDITIONAL EXPERIMENTAL ANALYSIS FOR BARLOW/SIMCLR + CASSLE

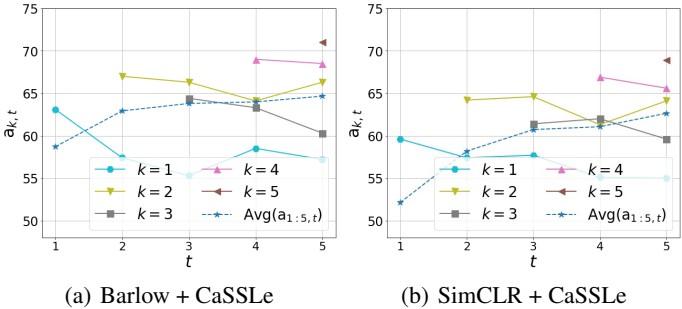

(a) Barlow + CaSSLe  (b) SimCLR + CaSSLe

Figure 6: The graph illustrates the values of $a_{k,t}$ for each algorithm in the Class-IL (5T) scenario. The measured stability and plasticity for each method are as follows: (a) $(S, P) = (2.52, 2.8)$, (b) $(S, P) = (2.22, 1.95)$.

Figure 6 illustrates an additional experimental analysis of Barlow + CaSSLe and SimCLR + CaSSLe in Class-IL (5T) experiments using the ImageNet-100 dataset. Similar to previous analyses, we once again observe an increase in their $\text{Avg}(a_{1:5,t})$; however, some tasks experience catastrophic forgetting. For instance, Barlow + CaSSLe exhibits a decline in performance in $a_{k=1,t}$ and $a_{k=3,t}$. Likewise, $a_{k=1,t}$ and $a_{k=4,t}$ of SimCLR + CaSSLe display a similar trend.

## B.5 ADDITIONAL EXPERIMENTAL RESULTS WITH OTHER METHODS

Given that results for prominent contrastive learning-based algorithms (SimCLR, MoCo V2) were already proposed, we introduce Supervised Contrastive Learning(SupCon) Khosla et al. (2020) as an additional algorithm to investigate the effectiveness of AugNeg in scenarios where labels are available. To accomplish this, we modified the implementation of AugNeg(SimCLR) in Equation (8)

and (9) adapting it to utilize supervised labels. More specifically, it incorporates additional negatives in SupCon + CaSSLe.

For the Non-contrastive learning-based algorithm, we selected the recent VICReg Bardes et al. (2022) algorithm to explore the potential application of AugNeg to various SSL algorithms. To implement AugNeg, we added additional regularization proposed in Equation (12) and conducted experiments accordingly.

We conducted experiments with the CIFAR-100 and ImageNet-100 datasets, employing the Class-IL (5T, 10T) scenario as our experimental setting, and the results are as follows.

Table 6: Experimental results for three downstream tasks.

| $A_T$ | | CIFAR-100 | | ImageNet-100 | |
|---|---|---|---|---|---|
| | | Class-IL 5T | Class-IL 10T | Class-IL 5T | Class-IL 10T |
| SupCon | CaSSLe | 60.38 | 55.38 | 66.26 | 60.48 |
| | **AugNeg** | **60.73** | **56.16** | **66.96** | **60.95** |
| VICReg | CaSSLe | 53.17 | 47.54 | 59.18 | 49.00 |
| | **AugNeg** | **55.40** | **50.78** | **61.88** | **51.82** |

Based on the results of the above experiments, we observe that AugNeg can be applied to additional contrastive and non-contrastive learning algorithms, leading to consistent performance improvement compared to CaSSLe. In particular, we observed successful application of our proposed AugNeg to VICReg, one of the latest SSL algorithms, achieving consistent performance improvements of 2-3% in both CIFAR-100 and ImageNet-100 experiments. We believe that these experimental findings suggest the potential for the proposed idea of AugNeg to be widely applicable to future various contrastive and non-contrastive learning algorithms.

## B.6 EVALUATION FOR DOWNSTREAM TASKS

Table 7: Experimental results for three downstream tasks.

| Scenario | | Downstream | MoCo + CaSSLe | SimCLR + CaSSLe | Barlow + CaSSLe | BYOL + CaSSLe | AugNeg (MoCo) | AugNeg (BYOL) |
|---|---|---|---|---|---|---|---|---|
| Class-IL | 5T | CIFAR-100 | 44.52 | 41.99 | 43.59 | 46.71 | 47.8 | 48.8 |
| | | CIFAR-10 | 68.09 | 64.9 | 65.43 | 69.42 | 71.26 | 70.21 |
| | | STL-10 | 63.23 | 63.3 | 65.39 | 67.93 | 67.07 | 68.59 |
| | | **Average** | **58.61** | **56.73** | **58.13** | **61.35** | **62.04** | **62.53** |
| | 10T | CIFAR-100 | 40.84 | 40.34 | 40.67 | 45.49 | 45.86 | 48.63 |
| | | CIFAR-10 | 66.01 | 64.62 | 64.42 | 68.02 | 68.65 | 70.34 |
| | | STL-10 | 60.16 | 59.77 | 62.38 | 65.55 | 65.78 | 67.51 |
| | | **Average** | **55.67** | **54.91** | **55.82** | **59.69** | **60.10** | **62.16** |
| Data-IL | 5T | CIFAR-100 | 44.5 | 42.33 | 44.5 | 47.02 | 47.07 | 48.14 |
| | | CIFAR-10 | 68.47 | 65.03 | 66.92 | 69.14 | 70.01 | 69.03 |
| | | STL-10 | 64.49 | 63.54 | 67.2 | 68.66 | 69.98 | 68.68 |
| | | **Average** | **59.15** | **56.97** | **59.54** | **61.61** | **62.35** | **61.95** |
| | 10T | CIFAR-100 | 42.17 | 41.3 | 43.53 | 45.3 | 46.88 | 48.05 |
| | | CIFAR-10 | 66.66 | 65.9 | 65.38 | 69.27 | 69.84 | 70.80 |
| | | STL-10 | 60.35 | 61.88 | 64.51 | 67.35 | 67.48 | 67.95 |
| | | **Average** | **56.39** | **56.36** | **57.81** | **60.64** | **61.40** | **62.27** |

As emphasized in multiple representation learning papers (Cha et al., 2023; Chen et al., 2020a; He et al., 2020), evaluating the generalization of learned representations across diverse downstream tasks is crucial. In line with this, we conducted evaluations on models trained with each CSSL scenario on the ImageNet-100 dataset, encompassing a range of downstream tasks. Following the methodology outlined in (Cha et al., 2023), we employed the resized CIFAR-10/-100 (Krizhevsky et al., 2009) datasets (resized to 96x96) and the STL-10 (Coates et al., 2011) dataset as downstream tasks and performed linear evaluations on them. The results, presented in Table 7, showcase the exceptional performance of AugNeg not only in the linear evaluation on the ImageNet-100 dataset (an in-domain dataset, as depicted in the manuscript) but also across various out-domain downstream task datasets, consistently achieving the best overall results. Furthermore, the proposed AugNeg exhibits superior representation learning in CSSL compared to other CaSSLe variations, particularly evident in the Data-IL scenario.

Table 8: Experimental results for CilPart in DomainNet (Peng et al., 2019)

| Scenario | | MoCo + CaSSLe | SimCLR + CaSSLe | Barlow + CaSSLe | BYOL + CaSSLe | **AugNeg (MoCo)** | **AugNeg (BYOL)** |
|---|---|---|---|---|---|---|---|
| Class-IL | 5T | 28.32 | 34.68 | 37.42 | 38.98 | **38.86** | **41.57** |
| | 10T | 28.57 | 33.33 | 38.30 | 38.05 | **38.19** | **38.05** |
| Data-IL | 5T | 29.74 | 34.17 | 36.13 | 37.04 | **38.33** | **40.06** |
| | 10T | 32.81 | 35.53 | 38.45 | 35.91 | **40.45** | **35.91** |

Secondly, following a similar approach as in the CaSSLe paper (Fini et al., 2022), we employed models trained with each CSSL scenario on the ImageNet-100 dataset and conducted linear evaluation using the CilPart dataset from DomainNet as the downstream task, with the results shown in Table 8. In the scenario of Class-IL, we observed that models trained with Barlow + CaSSLe or BYOL + CaSSLe achieve superior performance among baselines. However, AugNeg(MoCo) and AugNeg(BYOL) also shows competitive or state-of-the-art performance, especially in AugNeg(BYOL) in Class-IL 5T. Particularly in the case of Data-IL, similar to the results obtained from previous downstream task experiments, we observed that AugNeg(MoCo) and AugNeg(BYOL) outperform other algorithms by a considerable margin, except for AugNeg(BYOL) in Data-IL 10T.

### B.7 EXPERIMENTAL RESULTS FOR VARIOUS DIMENSIONS OF MLP IN PROJ LAYER

The use of larger dimensions for the Projection (or Prediction) layer has been known to facilitate the learning of superior representations in the self-supervised learning Chen et al. (2020a;b). Therefore, we conducted experiments to examine the performance variations of the proposed AugNeg(MoCo) by altering the dimension of the Projection layer. It is important to note that, for this experiment, we maintained the MLP dimension of the Prediction layer for distillation in AugNeg(MoCo) as 2048 and only adjusted the dimension of the Projection layer.

Table 9: Experimental results for different dimensions of MLP for the Proj layer (CIFAR-100, 5T).

| $A_5$ | Proj layer dim = 128 | Proj layer dim = 256 | Proj layer dim = 512 | Proj layer dim = 1024 | **Proj layer dim = 2048 (default)** | Proj layer dim = 4096 | Proj layer dim = 8192 | Proj layer dim = 16384 |
|---|---|---|---|---|---|---|---|---|
| AugNeg | 58.58 | 59.61 | 60.04 | 61.29 | **62.36** | 62.75 | 63.05 | 63.02 |

Table 9 presents the experimental results for the Class-IL 5T scenario using CIFAR-100. Similar to single-task self-supervised learning, it was empirically observed that a larger MLP dimension enhances the quality of learned representations after CSSL. While the default dimension of 2048 reached a certain level of convergence in all experiments, increasing the dimension to 4096 or 8192 yielded an additional performance gain of approximately 0.4-0.7%. This finding demonstrates that, when considering the dimensions of the Proj and Pred (for BYOL only) layers used by BYOL on CIFAR-100 (as shown in Table 9), AugNeg(MoCo) can create a slightly larger performance gap compared to BYOL when the MLP dimension in the Proj layer is set to be similar to that of BYOL.

Table 10: Experimental results for different dimensions of MLP for the Proj layer.

| $A_T$ | | **Proj layer dim = 2048 (default)** | Proj layer dim = 4096 | Proj layer dim = 8192 |
|---|---|---|---|---|
| Class-IL | ImageNet-100 (5T) | 67.70 | **68.26** | 66.92 |
| | ImageNet-100 (10T) | **60.65** | 60.32 | 60.43 |
| Data-IL | ImageNet-100 (5T) | 70.22 | **70.32** | 70.24 |
| | ImageNet-100 (10T) | **67.38** | 67.23 | 67.32 |
| Domain-IL | DomainNet (6T) | 46.74 | 47.06 | **47.53** |

Table 10 displays the experimental results obtained by increasing the MLP dimension of the Proj layer in AugNeg(MoCo) for experiments conducted on ImageNet-100 and DomainNet. Similar to the previous CIFAR-100 experiments, it was observed that increasing the MLP dimension beyond the default value of 2048 did not result in significant performance differences in most scenarios. However, in the case of Class-IL 5T or Domain-IL, when the dimension of the Proj layer was increased (i.e., set to a dimension similar to that of BYOL), an additional performance improvement of approximately 0.5-0.8% was achieved, leading to a larger performance gap (in Class-IL 5T) or a reduced gap (in Domain-IL) compared to BYOL.

## B.8 VARIOUS SIZES OF QUEUE FOR AUGNEG(MOCO)

Table 11: Experimental results for various sizes of queue for AugNeg(MoCo).

| $A_5$ | Queue size = 256 | Queue size = 512 | Queue size = 1024 | Queue size = 2048 | Queue size = 4096 | Queue size = 8192 | Queue size = 16384 | Queue size = 32768 | Queue size = 65536 (default) | Queue size = 131072 |
|---|---|---|---|---|---|---|---|---|---|---|
| CIFAR-100 5T | 61.03 | 61.33 | 61.68 | 61.84 | 61.90 | 61.95 | 62.10 | 61.99 | **62.36** | 62.01 |

Lastly, the experimental results for various queue sizes in AugNeg are presented in Table 11. As stated in the manuscript, the proposed AugNeg loss utilizes negative samples obtained from both the current and previous models, making the number of available negative samples during training crucial for performance. The findings in this table provide empirical evidence of this phenomenon. Performance shows no significant difference when the queue size exceeds 8192. However, a notable decline in performance is observed when the queue size is reduced, particularly at 256 which aligns with the mini-batch size. In this instance, when comparing the results of AugNeg(MoCo) using the queue size of 256 (60.10) to those of SimCLR + CaSSLe (58.87) in Table 1 , we once again observe the advantages of incorporating additional negatives. Furthermore, the negatives stored in the queue employed by AugNeg(MoCo) contain the features of samples used as positives in the previous iteration. Consequently, these can be seen to play the role of replaying exemplars of features from previous iterations. This suggests that, much like the significant benefits observed in traditional continual learning research when utilizing exemplar memory, employing such a queue can contribute to achieving better results in CSSL.

In conclusion, we are confident that these experiments validate the effective integration of AugNeg(MoCo) with MoCo. This is achieved through the utilization of a queue, allowing the inclusion of a significant number of negative samples used as positives in previous iterations, irrespective of the mini-batch size.

## B.9 EXPERIMENTAL RESULTS USING DIFFERENT MINI-BATCH SIZE

Table 12: Experimental results ($A_5$) of ImageNet-100 (Class-IL 5T).

| MoCo + FT | MoCo + CaSSLe | SimCLR + FT | SimCLR + CaSSLe | Barlow + FT | Barlow + CaSSLe | BYOL + FT | BYOL + CaSSLe | AugNeg (MoCo) | AugNeg (BYOL) |
|---|---|---|---|---|---|---|---|---|---|
| 55.80 | 61.70 | 53.70 | 62.40 | 59.00 | 65.00 | 59.20 | 61.60 | 67.40 | 63.34 |

In order to reproduce the results of CaSSLe, we trained with a mini-batch size of 256 during the CSSL process, and then conducted the same linear evaluation as described in the main text, as shown in Table 12. From the experimental results, we found that increasing the mini-batch size made it challenging to faithfully reproduce the results of the CaSSLe paper, and, in fact, we obtained slightly lower results compared to the results mentioned in the manuscript (with a mini-batch size of 128). However, for AugNeg(MoCo) and AugNeg(BYOL), we observed similar results even across two different mini-batch size settings.

## B.10 TRAINING DETAILS FOR CONTINUAL SELF-SUPERVISED LEARNING

Table 13 presents the training details for each algorithm utilized in the Continual Self-supervised Learning (CSSL) experiments. All experiments were run on an NVIDIA RTX A5000 with CUDA 11.2. We employed LARS (You et al., 2017) to train an encoder in the scenario of CSSL. It is worth emphasizing that we strictly followed the training settings of BarlowTwins, as implemented in CaSSLe, including crucial factors such as learning rate, weight decay, learning rate schedule, and augmentations. Notably, we did not perform an extensive search for hyperparameters.

It is important to mention that all self-supervised learning algorithms use a Projection layer Chen et al. (2020a), consisting of two layers of MLP with ReLU activation. Specifically, BYOL incorporates a larger default dimension for the Projection layer (dim = 4096) and includes its own Prediction layer (for BYOL) that is not utilized by other self-supervised learning algorithms. This indicates that BYOL assigns a larger portion of weights to the MLP layer in the CSSL process. The subsequent sections will present the consistent results of AugNeg, accounting for these factors.

For the Domain-incremental learning (Domain-IL) experiments conducted on the DomainNet dataset, we followed the experimental setup described in the CaSSLe paper Fini et al. (2022). Specifically,

Table 13: Experimental details for CSSL. (CIFAR-100 / ImageNet-100 / DomainNet)

| CIFAR-100 / ImageNet-100 / DomainNet | MoCo (+CaSSLe) | SimCLR (+CaSSLe) | BarLow (+CaSSLe) | BYOL (+CaSSLe) | AugNeg (MoCo/BYOL) |
|---|---|---|---|---|---|
| Epoch (per task) | 500 / 400 / 200 | | | | |
| Batch size | 256 / 128 / 128 | | | | |
| Learning rate | 0.4 | 0.4 | 0.3 / 0.4 / 0.4 | 1.0 / 0.6 / 0.6 | 0.6 |
| Optimizer | SGD | | | | |
| Weight decay | 1e-4 | 1e-4 | 1e-4 | 1e-5 | 1e-4 (for MoCos) 1e-5 (for BYOL) |
| Projection layer (dim) | 2048 | 2048 | 2048 | 4096 | 2048 |
| Prediction layer (dim, for CSSL) | 2048 | | | | |
| Prediction layer (dim, for BYOL) | - | - | - | 4096 / 8192 / 8192 | 4096 / 8192 / 8192 (for BYOL only) |
| Queue | 65536 | - | - | - | 65536 (for MoCo only) |
| Temperature ($\tau$) | 0.2 | 0.2 | - | - | 0.2 (for MoCo only) |

we performed Domain-IL in the order of Real => QuickDraw => Painting => Sketch => InfoGraph => CilPart.

### B.11 OTHER EXPERIMENTAL DETAILS

**Linear evaluation** Table 14 provides the training details used for linear evaluation on each dataset using the encoders trained through CSSL. In the case of CaSSLe variations, we conducted the experiments while keeping the linear evaluation settings consistent with those used in CaSSLe.

Table 14: Experimental details for lienar evaluation. (CIFAR-100 / ImageNet-100 / DomainNet)

| CIFAR-100 / ImageNet-100 / DomainNet | MoCo (+CaSSLe) | SimCLR (+CaSSLe) | BarLow (+CaSSLe) | BYOL (+CaSSLe) | AugNeg |
|---|---|---|---|---|---|
| Epoch (per task) | 100 | | | | |
| Batch size | 128 / 256 / 256 | | | | |
| Learning rate | 3.0 | 1.0 | 0.1 | 3.0 | 3.0 |
| Scheduler | Step LR (steps = [60, 80], gamma = 0.1) | | | | |
| Optimizer | SGD | | | | |
| Weight decay | 0 | | | | |

**Downstream task** Table 15 presents the settings utilized for linear evaluation on downstream tasks using models trained with each CSSL scenario on the ImageNet-100 dataset. In line with the algorithms based on CaSSLe, we adopted the experimental settings outlined in the respective paper. As a result, we observed that distinct learning rates were employed for linear evaluation across different algorithms, highlighting variations in their downstream task assessments. Likewise, we conducted experiments for AugNeg with an optimized learning rate to present the most favorable results.

Table 15: Experimental details for linear evaluation on downstream tasks.

| CIFAR-10 / CIFAR-100 / STL-10 / CilPart | MoCo (+CaSSLe) | SimCLR (+CaSSLe) | BarLow (+CaSSLe) | BYOL (+CaSSLe) | AugNeg |
|---|---|---|---|---|---|
| Epoch (per task) | 100 | | | | |
| Batch size | 256 | | | | |
| Learning rate | 3.0 | 1.0 | 0.1 | 3.0 | 3.0 |
| Scheduler | Step LR (steps = [60, 80], gamma = 0.1) | | | | |
| Optimizer | SGD | | | | |
| Weight decay | 0 | | | | |

# C    IMPLEMENTATION DETAILS

## C.1    PSEUDO CODE OF AUGNEG WITH SSL ALGORITHMS

In this section, we provide a detailed explanation of the practical implementation of the loss function described in Equation (2) of the manuscript. As discussed in Section 3, the proposed AugNeg loss can be easily implemented for SimCLR (Chen et al., 2020a) by utilizing 2N augmented images from the input mini-batch. However, for MoCo v2 Plus (Chen et al., 2020b), which employs a queue, and BarlowTwins (Zbontar et al., 2021), which relies on a loss function based on the cross-correlation matrix, additional explanations are required to implement AugNeg loss using these approaches. To address this, we provide pseudo code for implementing AugNeg loss based on MoCo and BarlowTwins in Algorithms 1 and 2, respectively. For more detailed implementation instructions, please refer to the accompanying source code.

**Algorithm 1** AugNeg with MoCo v2 Plus Chen et al. (2020b).

1: **function** AUGNEG(query1, key1, queue1, distill1, query2, queue2, temp)
2:     l_pos1 = einsum('nc,nc->n', [query1, key1]).unsqueeze(-1)
3:     l_neg1 = einsum('nc,ck->nk', [query1, cat([queue1, queue2], dim = 1)])
4:     logits1 = cat([l_pos1, l_neg1], dim=1)
5:     logits1 /= temp
6:     labels = zeros(logits1.shape[0]).cuda()
7:     loss1 = cross_entropy(logits1, labels)
8:
9:     l_pos2 = einsum('nc,nc->n', [distill1, query2]).unsqueeze(-1)
10:     l_neg2 = einsum('nc,ck->nk', [distill1, cat([queue1, queue2], dim = 1)])
11:     logits2 = cat([l_pos2, l_neg2], dim=1)
12:     logits2 /= temp
13:     loss2 = cross_entropy(logits2, labels)
14:
15:     loss = loss1 + loss2
16:     return loss
17: **end function**
**Require:**
18: $h_{\theta_t}$, $h_{\theta_t}^m$: $t$-th task's encoder and momentum network
19: $h_{\theta_{t-1}}$: encoder network for the previous task.
20: $\mathcal{Q}_t^m$, $\mathcal{Q}_{t-1}$: queues.
21: $m$: momentum.
22: $\tau$: temperature.
**Ensure:** $h_{\theta_t}$: encoder after learning the task $t$.
23: # Starting from the end of learning $t-1$ (previous) task.
24: $h_{\theta_t} \leftarrow h_{\theta_{t-1}}$     # initialize task $t$-th's encoder.
25: $h_{\theta_t}^m \leftarrow h_{\theta_{t-1}}$     # initialize task $t$-th's momentum encoder.
26: $Proj_t \leftarrow Proj_{t-1}$     # initialize Proj layer of $h_{\theta_t}$.
27: $Proj_t^m \leftarrow Proj_{t-1}$     # initialize Proj layer of $h_{\theta_t}^m$.
28: $\mathcal{Q}_t^m \leftarrow \mathcal{Q}_{t-1}^m$     # initialize $\mathcal{Q}_t^m$.
29: $Pred, \mathcal{Q}_{t-1} \leftarrow RandInit$     # initialize Pred layer with random initialization.
30:
31: **for** $x$ **in** loader **do**     # load a minibatch x with N samples.
32:     $x^a, x^b = Aug^a(x), Aug^b(x)$     # applying augmentations
33:
34:     $q_t^a, q_t^b = Norm(Proj_t(h_{\theta_t}(x^a))), Norm(Proj_t(h_{\theta_t}(x^b)))$
35:     $d_t^a, d_t^b = Norm(Pred(q_t^a)), Norm(Pred(q_t^b))$
36:     $k_t^a, k_t^b = Norm(Proj_t^m(h_{\theta_t}^m(x^a))).detach(), Norm(Proj_t^m(h_{\theta_t}^m(x^b))).detach()$
37:
38:     $q_{t-1}^a, q_{t-1}^b = Norm(Proj_{t-1}(h_{\theta_{t-1}}(x^a))).detach(), Norm(Proj_{t-1}(h_{\theta_{t-1}}(x^b))).detach()$
39:
40:     loss = AugNeg($q_t^a, k_t^b, \mathcal{Q}_t^m, d_t^a, q_{t-1}^a, \mathcal{Q}_{t-1}, \tau$) + AugNeg($q_t^b, k_t^a, \mathcal{Q}_t^m, d_t^b, q_{t-1}^b, \mathcal{Q}_{t-1}, \tau$)
41:     (loss*0.5).backward(), update($h_{\theta_t}, Proj_t, Pred$)
42:
43:     enqueue($\mathcal{Q}_t^m$, cat($k_t^a, k_t^b$)), enqueue($\mathcal{Q}_{t-1}$, cat($q_{t-1}^a, q_{t-1}^b$))     # enqueue the current batch
44:     dequeue($\mathcal{Q}_t^m$), dequeue($\mathcal{Q}_{t-1}$)
45:     # momentum update using $t$-th model.
46:     momentum_update($h_{\theta_t}, h_{\theta_t}^m, m$), momentum_update($Proj_t, Proj_t^m, m$)
47: **end for**
48: # The end of learning $t$ (current) task.

---

**Algorithm 1** AugNeg with BarlowTwins Zbontar et al. (2021).

---

1: **function** AUGNEG(z1, z2, z1_prev, z2_prev, z1_distill, z2_distill, $\lambda = 0.005$, scale= 0.1)
2:
3:      N, D = z1.size()
4:      bn = BatchNorm1d(D, affine=False).to(z1.device)
5:
6:      z1 = bn(z1)
7:      z2 = bn(z2)
8:      z1_prev = bn(z1_prev)
9:      z2_prev = bn(z2_prev)
10:     z1_distill = bn(z1_distill)
11:     z2_distill = bn(z2_distill)
12:
13:     corr1 = einsum(bi, bj -> ij, z1, z2) / N
14:     corr1_prev = einsum(bi, bj -> ij, z1, z2_prev) / N
15:     corr2 = einsum(bi, bj -> ij, z1_distill, z1_prev) / N
16:     corr2_prev = einsum(bi, bj -> ij, z1_distill, z2) / N
17:
18:     diag = eye(D)
19:     cdif1 = (corr1 - diag).pow(2)
20:     cdif1[ diag.bool()] *= $\lambda$
21:     # off_diagonal: get off diagonal values of a given matrix.
22:     off_diag_prev1 = off_diagonal(corr1_prev).pow(2).sum()
23:     loss1 = scale * (cdif1.sum() + off_diag_prev1 * $\lambda$)
24:
25:     cdif2 = (corr2 - diag).pow(2)
26:     cdif2[ diag.bool()] *= $\lambda$
27:     off_diag_prev2 = off_diagonal(corr2_prev).pow(2).sum()
28:     loss2 = scale * (cdif2.sum() + off_diag_prev2 * $\lambda$)
29:
30:     loss = (loss1 + loss2)
31:     return loss
32: **end function**
**Require:**
33: $h_{\theta_t}, h_{\theta_t}^m$: $t$-th task's encoder and momentum network
**Ensure:** $h_{\theta_t}$: encoder after learning the task $t$.
34: # Starting from the end of learning $t-1$ (previous) task.
35: $h_{\theta_t} \leftarrow h_{\theta_{t-1}}$      # initialize task $t$-th's encoder.
36: $Proj_t \leftarrow Proj_{t-1}$      # initialize Proj layer of $h_{\theta_t}$.
37: $Proj_t^m \leftarrow Proj_{t-1}$      # initialize Proj layer of $h_{\theta_t}^m$.
38: $Pred \leftarrow RandInit$      # initialize Pred layer with random initialization.
39:
40: **for** $x$ **in** loader **do**      # load a minibatch x with N samples.
41:     $x^a, x^b = Aug^a(x), Aug^b(x)$      # applying augmentations
42:
43:     $q_t^a, q_t^b = Norm(Proj_t(h_{\theta_t}(x^a))), Norm(Proj_t(h_{\theta_t}(x^b)))$
44:     $d_t^a, d_t^b = Norm(Pred(q_t^a)), Norm(Pred(q_t^b))$
45:
46:     $q_{t-1}^a, q_{t-1}^b = Norm(Proj_{t-1}(h_{\theta_{t-1}}(x^a))).detach(), Norm(Proj_{t-1}(h_{\theta_{t-1}}(x^b))).detach()$
47:
48:     loss = AugNeg($q_t^a, q_t^b, q_{t-1}^a, q_{t-1}^b, d_t^a, d_t^b$) + AugNeg($q_t^b, q_t^a, q_{t-1}^b, q_{t-1}^a, d_t^b, d_t^a$)
49:     (loss*0.5).backward(), update($h_{\theta_t}, Proj_t, Pred$)
50:
51: **end for**
52: # The end of learning $t$ (current) task.

---

