# OpenReview forum: "Augmenting Negative Representation for Continual Self-Supervised Learning"
_ICLR.cc/2024/Conference — Submitted to ICLR 2024_

### Official Review · Reviewer_GiPi · 2023-10-20

**Soundness:** 3 good
**Presentation:** 3 good
**Contribution:** 3 good
**Rating:** 6
**Confidence:** 3

**Summary:**

In this study, the authors introduce a novel method for continual self-supervised learning, termed "AugNeg". This approach generates an increased number of negative examples, utilizing the encoding derived from the previous model. Demonstrating versatility, the proposed method exhibits enhancements across three distinct settings. Furthermore, the authors integrate this method with non-contrastive learning methods, adapting it into a regularization term.

**Strengths:**

1. Continual self-supervised learning stands as a promising field of research, offering the substantial benefit of potentially reducing computational resource requirements.
2. The proposed method appears to be soundness. By generating a greater number of negative examples, particularly those derived from the previous model, it is anticipated that the quality of the representations will be enhanced.
3. The structure of this paper is commendably clear and logical, facilitating ease of understanding and follow-through for readers.

**Weaknesses:**

1. There is a significant discrepancy in performance between the proposed method and standard joint training. It raises the question of whether the proposed method offers any resource savings compared to conventional training approaches. Additionally, it's pertinent to question why a user would choose to sample exclusively from current task data instead of the entire dataset.

2. The primary goal of continual self-supervised learning (CSSL) is to conserve resources, for instance, by reducing the need for large mini-batch sizes. However, it's crucial to determine whether the proposed method maintains its efficacy in datasets with an extensive array of classes, such as ImageNet-1K.

3. Augmentation represents a crucial area of exploration in Self-supervised learning. Given that the authors classify their method as a form of augmentation, it becomes essential to engage in comparisons and discussions with existing augmentation methods [1][2][3].

[1] ReSSL: Relational Self-Supervised Learning with Weak Augmentation, NeurIPS 2021. \
[2] RSA: Reducing Semantic Shift from Aggressive Augmentations for Self-supervised Learning, NeurIPS 2022. \
[3] Masked Autoencoders Are Scalable Vision Learners, CVPR 2022.

**Questions:**

1. In Figure 2, z+,i,t-1 is regarded as an negative example. Is it a typo?  Additionally, with new negative examples, the gradient looks keeping nearly the same direction.

2. Equation 5 is still unclear and requires further elaboration. It looks offsetting to Equation 4.

---

> ### Author Response · Authors · 2023-11-17
> **Author response 1**
>
> **Weakness 1: The scenario of continual self-supervised learning and advantages of CSSL in resource saving**
>
> Thank you for your insightful question. To provide a detailed response to the reviewer's inquiry, we will reiterate the setting of continual self-supervised learning (CSSL) outlined in Section 3.1 and Figure 1(b). As depicted in Figure 1(b), the CSSL scenario considers a situation where the training dataset is incrementally added. Specifically, it allows sequential access to each task dataset ($D\_t$) during the training of the $t$-th task. **However, it explicitly acknowledges the impossibility of accessing previous task datasets ($D\_{1, ..., t-1}$).** In this context, the challenge arises from the unavailability of data sampling from the entire dataset ($D\_{1, ..., t}$), leading to the Low Diversity of Negative Samples problem, a key motivation in CSSL as discussed in Section 3.2 and illustrated in Figure 1(b).
>
> Training with the CSSL scenario has two benefits: **1) reducing the demand for memory resource and 2) minimizing GPU cost for training.** In the same CSSL scenario mentioned earlier, consistently training on the entirety of data (Joint) can be considered a conventional training approach. Consequently, when the size of each task dataset is constant (e.g., $\|D\_t\| = N$), retraining the Joint model at each $t$ task incurs linear increases in both memory and GPU resources (e.g., $N\*t$). Considering the billing structure of major cloud services (e.g., AWS) based on resource usage over a unit period, the total cost for memory and GPU resources until obtaining the $t$-th Joint model is approximately $N\*(t+1)\*t/2$. In contrast, for CSSL, the total memory cost is $N\*t$ to obtain the $t$-th model since it uses only the new $t$-th task dataset for each task $t$. Regarding GPU resources, with an additional cost $\alpha$ introduced by the CSSL algorithm, the approximate cost becomes $N + (N + \alpha)\*(t-1)$.
>
> In summary, when considering the cost of memory and GPU resources in each learning scenario, **CSSL algorithms offer a significant cost reduction compared to the Joint**. Furthermore, the cost efficiency of CSSL algorithms becomes more pronounced as the size of each task's training dataset increases (i.e., as N increases). Taking this cost efficiency into account, we believe that **achieving performance levels of the proposed AugNeg compared to the Joint, from approximately 93% (AugNeg(MoCo) on CIFAR-100, Class-IL 5T) to 96% (AugNeg(BYOL) on DomainNet, Domain-IL 6T), is a substantial and meaningful outcome.**
>
>
> **Weakness 2: Additional experiments in ImageNet-1k**
>
> We agree that presenting additional experimental results on ImageNet-1k would contribute significantly to validating the efficiency of our algorithm. We are currently conducting them as supplementary experiments. However, given our current limited GPU resources, obtaining the complete set of experimental results may take considerable time. If we manage to complete any partial results before the end of the author response period, we commit to sharing those results.
>
>
> **Weakness 3: Additional related works on augmentation in SSL**
>
> Thank you for sharing valuable papers. Upon comparing our paper with the ones you provided, **we believe our distinctive contribution lies in proposing to consider additional negative samples obtained from both the current and previous models in Continual Self-Supervised Learning (CSSL) scenarios**. From this perspective, we acknowledge the importance of comparing and discussing our method with those designed for successful self-supervised learning in the conventional single-task setting. In a forthcoming revision, we will update the related work section to incorporate discussions on augmentation methods including the papers the reviewer shared.

---

> ### Author Response · Authors · 2023-11-17
> **Author response 2**
>
> **Q1: Figure 2 and the gradient of the proposed AugNeg loss**
>
> We apologize for any confusion caused by the unclear Figure 2. We have revised the figure to better align with the explanation in Equation 2.
> Additionally, we responded the reviewer's question on the gradient in **Common Response 3**.
>
> **Q2: Additional explanation on Equation 5**
>
> we responded the reviewer's question on Equation 5 in **Common Response 4**.

---

> > ### Comment · Reviewer_GiPi · 2023-11-21
> > **Thanks for the thoughtful response.**
> >
> > Most of my concerns have been adequately addressed, leading me to increase my evaluation score accordingly.

---

> > > ### Author Response · Authors · 2023-11-21
> > > **Thanks for the reviewer's response!**
> > >
> > > Thank you for responding to our response! We appreciate the reviewer GiPi's reconsideration of the evaluation. We will make sure to submit a revised version that incorporates our responses before the end of the discussion period. Thanks once again.

---

### Official Review · Reviewer_Tkp5 · 2023-10-30

**Soundness:** 2 fair
**Presentation:** 2 fair
**Contribution:** 2 fair
**Rating:** 5
**Confidence:** 3

**Summary:**

This work present the improvement to SSL loss function for continual self-supervised learning (CSSL) that consider outputs of the previous model while training for the current task. The proposed loss consists of two terms: plasticity and stability ones, without additional explicit tradeoff between two of them. Proposed method should result in a better plasticity in comparison to existing method, that focus on stability. Experimental section follows one from CaSSLe method [18]. Ablation study is provided.

**Strengths:**

1. Motivation why use negatives from the previous tasks is well motivated.
1. Proposed method presents some improvement.

**Weaknesses:**

The main weakness for me of this paper is seeing it for the second time without small changes. I've spent some of my time to help the authors to improve it the last time and they do not even find enough time to do the good text replacement from Sy-con to AugNeg.
Main changes: They've removed one unfavorable plot (CaSSLe was always better on it for two out of three methods) and add Table 2, changed Fig.4, added "SyCON (BYOL)" **(!) (authors own writting)**, and added why the cannot reproduce CaSSLe results (Challenges of reproducing CaSSLe’s results - I've checked the issue page as well, 2% changed at the end in #12).

1. Improvements presented in Table 1 (CSSL baselines) – taking into account the results variability is not always significant (see std. dev. reported there for AugNeg vs CaSSLe).

1. The results for CaSSLe are still lower from ones presented in the original paper. The pointed issue on the github is mainly about BYOL method.

1. For a good comparison in Fig.4 the right figure should be MoCo with just finetuning. We can then compare all three methods better and can be a good sanity check. Right now, what we can say is that AugNeg hurts the performance on the first task a bit (why?) and is better in the following. Do we have the same queue size and all hyper-params here between the methods? (see my questions).

1. There is no clear message why AugNeg works in each of the scenario with each method (MoCo / BYOL).

**Questions:**

1. Why do not adjust other hyper-parameters, when changing some crucial ones, e.g. batch-size for cassle?

1. Why AugNeg for Domain-IL is 43 (Tab.3) when SyCON for the same seting was 46?

1. Is the MoCo for FT and CaSSLe as well run with extended queue size (to 65K)?

---

> ### Author Response · Authors · 2023-11-17
> **Author response 1**
>
> **Weakness 0: Changes from the previous submission version**
>
> Firstly, we regret that the extent of our paper's updates might not meet the reviewer's satisfaction. We appreciate all the feedback received so far and have made diligent efforts to enhance the paper based on the provided comments. The updates in this submission include:
>
> 1. **Title and Algorithm Names:** revised the paper title and the names of the proposed algorithm.
> 2. **Introduction Enhancement:** Improved the final paragraph in the Introduction.
> 3. **Method Section:**
>    - Attenuated the discussion on an issue with conventional loss function and introduced a discussion on the problem of the low diversity of negatives.
>    - Newly introduced the idea of extending AugNeg to non-contrastive learning in Section 3.4. Analytical details on existing gradients and other specifics have been relocated to the Supplementary Material.
>
> 4. **Experimental Section Changes:**
>    - Re-performed AugNeg experiments on CIFAR-100, with results reported in Figure 3 and Table 1.
>    - Tried to address the reproduce issue of CaSSLe in Table 3, essentially re-conducting the main experiments. As a result, achieved slightly improved results consistently compared to the previous version.
>    - Added more experimental scenarios for downstream task.
>    - Included experiments in the Supplementary Materials covering the size of the Proj(MLP) layer, results based on the queue size in AugNeg(MoCo), and experiments with an increased mini-batch size.
>
> Also, we will further enhance our paper by incorporating the comments received in this review through additional revisions.

---

> ### Author Response · Authors · 2023-11-17
> **Author response 2**
>
> **Weakness 1: Improvements presented in Table 1 is not significent**
>
> While the numeric improvement of AugNeg over CaSSLe may appear modest, we believe it gains a competitive edge when considering performance of Joint. For example, AugNeg(MoCo) shows a relatively modest increase of 3.6% over MoCo + CaSSLe. However, considering that MoCo + CaSSLe already achieves performance close to the upper bound (Joint) at around 89.8%, the fact that AugNeg(MoCo) reaches the Joint's performance of 93.2% with a similar training cost and low standard deviation is deemed a meaningful improvement. Moreover, we assert that our AugNeg variants demonstrate superior performance in other experiments, including ImageNet-100, DomainNet, and downstream tasks. We would appreciate the reviewer considering these points.
>
> **Weakness 2: Reproduce issue of CaSSLe**
>
> We made diligent efforts to reproduce the CaSSLe experiments in the manuscript, but unfortunately, we were unable to achieve results identical to those reported in the CaSSLe paper. With the Github issue #12 concerning BYOL in the ImageNet-100 experiment, there is  issue #10 discussing reproducibility concerns in the ImageNet-100 experiment. Note both indicate a common trend of obtaining results 5-6% lower than those reported in the original paper, aligning with the tendencies observed in our experimental results. Also, these issues remain unresolved, and the authors have not provided a comprehensive solution.
>
> We hope the reviewer has taken into account the current challenging situation regarding reproducibility. Additionally, we conducted all experiments of our paper in the unified environment and have made the experimental code publicly available. We would appreciate it if these considerations could be taken into account in the final evaluation.
>
>
> **Weakness 3: Suggestion for Figure 4**
>
> Thank you for your valuable feedback. We have updated Figure 4 in the manuscript and made corresponding modifications according to the reviewer's comments. Additionally, in the Supplementary Materials, we included the same analysis on AugNeg(BYOL), so please take a look. The discrepancy in Task 0 performance between MoCo + CaSSLe and AugNeg(MoCo) in the original graph was due to a typo. As evident in Figure 3, for all experiments in our paper, we initiated training with the same Task 0 model, conducting both CaSSLe and AugNeg, and reported the results accordingly. We apologize for the typo and have corrected it.
>
>
> **Weakness 4: No clear message**
>
> See Common Response 3 and 4. We will update our paper following these responses.

---

> ### Author Response · Authors · 2023-11-17
> **Author response 3**
>
> **Q1: Experimental results using other hyperparameters**
>
> We already reported experimental results on various dimensions of MLP, various queue sizes, and large mini-batch size in the Supplementary Materials.
>
> **Q2: Experimental result on Domain-IL**
>
> As mentioned before, we have re-conducted most of the experiments while writing the current paper, so we redrew the figures from scratch. Upon reviewing the previous version of the paper, it seems there was a typo in the table. We apologize for this typo. We have double-checked all results, and there are no typos in the currently drawn tables and figures.
>
> **Q3: Experimental result of MoCo with a extended queue size**
>
> Firstly, in this paper, the MoCo experiments used a default queue size of 65,536. We believe the reviewer's intention is to inquire about the results when using a queue of size 65,536 $\times$ 2 in MoCo + CaSSLe (please clarify if otherwise). The results for this experiment are already presented as one of the ablation studies in Case 4 of Table 5 in the manuscript. From this experiment, we experimentally validated that using additional negatives is distinct from simply increasing the queue size, as MoCo + CaSSLe with a doubled queue size (60,09) shows little difference compared to the original CaSSLe results (60.11).

---

> ### Author Response · Authors · 2023-11-22
> **Author response 4**
>
> Dear Reviewer Tkp5,
>
> Thank you once again for providing insightful comments. We have made every effort to respond to all your comments within the limited discussion period and have completed the second revision, incorporating all reviewer’s comment into the manuscript.
>
> We want to reiterate that while our proposed AugNeg is a simple idea, its core contribution lies in identifying issues from a different perspective in the field of CSSL research (specifically, pointing out the problem of both 1) the conventional loss form and 2) the low diversity of negative samples). We have introduced a novel loss function to overcome these challenges via model-based augmentation of negative representations. Additionally, through diverse experiments using not only CIFAR-100 but also ImageNet-100, DomainNet and downstream tasks, we experimentally demonstrated that AugNeg achieves consistent performance improvement with little difference in training cost compared to CaSSLe. Furthermore, we believe that the analysis suggested by the reviewer clearly highlights the advantages of AugNeg, confirming its ability to achieve enhanced both plasticity and stability, as aligned with the Method Section.
>
> As the discussion period is coming to a close, we would greatly appreciate it if the reviewer could review our response (and ideally the revised manuscript) and provide feedback. Thank you again.

---

### Official Review · Reviewer_E4AJ · 2023-11-01

**Soundness:** 3 good
**Presentation:** 2 fair
**Contribution:** 2 fair
**Rating:** 5
**Confidence:** 4

**Summary:**

The paper introduces Augmented Negatives (AugNeg), a new approach for continual self-supervised learning (CSSL). It addresses limitations in the conventional loss function by incorporating two losses that balance plasticity and stability. The authors evaluate AugNeg's performance over existing methods in both contrastive and non-contrastive CSSL on CIFAR-100, ImageNet100, and DomainNet on Class-, Data-, and Domain-IL.

**Strengths:**

The author thoroughly re-examine the drawbacks of the existing algorithm (CaSSLe) and propose a novel loss function.

**Weaknesses:**

1. More experimental results are expected. CaSSLe performs experiments with all SSL methods mentioned in all CSSL settings. However, the authors only selected two SSL methods, though with exploratory experiments on CIFAR-100, to compare with the baselines. It is worth noting that different SSL methods may have different effects on different CSSL settings and datasets. The goal of various SSL methods is to demonstrate that the loss can universally improve CSSL, given any existing SSL methods and potentially future methods.
2. The presentation of the paper needs to be improved, as most of the captions of the tables and figures do not contain useful information.
3. The loss needs a more intuitive explanation. From the current presentation, it seems like the design lacks meaning and is more of an engineer labor. See the question below.

**Questions:**

While I can understand that additional embeddings $\mathcal{N}_{t-1}(i)$ are introduced to $\mathcal{L}_1$, I am curious about the effect of this operation in the representation space and the specific property that $\mathcal{L}_1$ aims to encourage. Are the current negative samples in $\mathcal{L}1$ (previous negative samples in $\mathcal{L}2$) so extensively utilized that the inclusion of the negative samples from another source is necessary? If this is indeed the case, could it be attributed to the scale of the dataset? The variable

$z_{i,t} = Proj(h_{\theta_t}(x_i))$

in CaSSLe is designed for the invariant feature of the current and previous tasks. Does this apply to the proposed algorithms as well? In section 3.4, why does the addition of an extra regularization term follow a similar principle to that of the previous section?

---

> ### Author Response · Authors · 2023-11-17
> **Author response 1**
>
> **Weakness 1: Applicability of AugNeg to other methods**
>
> We appreciate the valuable suggestion. **The rationale behind conducting experiments focused on MoCo V2, SimCLR, BarLow, and BYOL is rooted in the findings of the CaSSLe paper, where these algorithms demonstrated superior performance across the majority of datasets, with MoCo V2 and BYOL particularly excelling.** In alignment with the reviewer's suggestion, we conducted additional experiments by selecting one contrastive and one non-contrastive learning-based algorithm each to explore the applicability of AugNeg to various continual representation learning scenarios.
>
> Given that results for prominent contrastive learning-based algorithms (SimCLR, MoCo V2) were already presented in the manuscript, **we introduced Supervised Contrastive Learning(SupCon) [1] as an additional algorithm** to investigate the effectiveness of AugNeg in scenarios where labels are available. To accomplish this, we modified the implementation of SimCLR in Equation (2) of the manuscript, adapting it to utilize supervised labels, specifically incorporating additional negatives in SupCon + CaSSLe.
>
> For the Non-contrastive learning-based algorithm, **we selected the recent VICReg [2] algorithm** to explore the potential application of AugNeg to various SSL algorithms. To implement AugNeg, we added additional regularization proposed in Equation (5) of the manuscript and conducted experiments accordingly.
>
> We conducted experiments with the CIFAR-100 and ImageNet-100 datasets, employing the Class-IL (5T, 10T) scenario as our experimental setting, and the results are as follows.
>
> | Dataset (5T, ResNet-18) | Alg. | SupCon | VICReg |
> | -------- | -------- | -------- | -------- |
> | CIFAR-100 | CaSSLe | 60.38 | 53.17 |
> | CIFAR-100 | AugNeg | 60.73     |  55.40    |
> | ImageNet-100 | CaSSLe | 66.26 | 59.18 |
> | ImageNet-100 | AugNeg | 66.96 | 61.88     |
>
> | Dataset (10T, ResNet-18) | Alg. | SupCon | VICReg |
> | -------- | -------- | -------- | -------- |
> | CIFAR-100 | CaSSLe | 55.38 | 47.54 |
> | CIFAR-100 | AugNeg | 56.16     |  50.78    |
> | ImageNet-100 | CaSSLe | 60.48 | 49.00 |
> | ImageNet-100 | AugNeg | 60.96     |  51.82    |
>
>
> Based on the results of the above experiments, we observe that **AugNeg can be applied to additional contrastive and non-contrastive learning algorithms, leading to consistent performance improvement compared to CaSSLe**. In particular, we observed successful application of our proposed AugNeg to VICReg, one of the latest SSL algorithms, achieving consistent performance improvements of 2-3% in both CIFAR-100 and ImageNet-100 experiments. We believe that these experimental findings suggest the potential for the proposed AugNeg to be widely applicable to future various contrastive and non-contrastive learning algorithms.
>
> [1] Khosla, Prannay, et al. "Supervised contrastive learning." Proceedings of the 34th International Conference on Neural Information Processing Systems. 2020.
>
> [2] Bardes, Adrien, Jean Ponce, and Yann LeCun. "VICREG: VARIANCE-INVARIANCE-COVARIANCE REGULARIZATION FOR SELF-SUPERVISED LEARNING." 10th International Conference on Learning Representations, ICLR 2022. 2022.
>
> -- revision of the response --
>
> We discovered a error in the experiment with AugNeg(VICReg) on ImageNet-100. After rectifying the error and re-running the experiment, we have input the revised results into the table and adjusted the response accordingly. We apologize for any inconvenience caused by this.

---

> ### Author Response · Authors · 2023-11-17
> **Author response 2**
>
> **Weakness 2: Captions and Figures**
>
> We have revised and updated the captions for all Figures and Tables in the manuscript.
>
> **Weakness 3 and Q1) The more intuitive explanation for AugNeg Loss**
>
> Thank you for the question. We have summarized our response regarding the role of additional negatives of the proposed AugNeg Loss in Common Response 3, so please check that for details.
>
> Additionally, **we would like to clarify that $\mathcal{N_{t-1}}$ is not $\mathcal{N_{t}}$ employed in $\mathcal{L}_1$ during the previous task (t-1) training.** As stated in the manuscript, these refer to negatives obtained from $t$ and $t-1$ models during training task $t$. Their implementation varies based on the contrastive learning algorithm; for SimCLR, all samples in the current mini-batch, excluding the identical sample, are considered as negatives. While for MoCo, the entire queue containing negatives used in previous iterations as a positive serves as negatives for the current iteration. At each training iteration for each task, even with the same input mini-batch, the $t$ and $t-1$ models have distinct output features for negative samples (as the $t$ model continues to update, while the $t-1$ model remains frozen). The core idea behind the AugNeg Loss proposed in Equation (2) is to concurrently consider these diverse features in CSSL.
>
> **Q2) Projection layer used in CaSSLe**
>
> Indeed, we utilized it. However, in Section 3.3, $z_{i, t} = Proj(h_{\theta_t}(x_i))$ represents the standard Projection layer commonly employed in SSL. We defined the Predictor for CSSL as $\tilde{z}\_{i, t} = Pred(z\_{i,t})$ and employed them to express the AugNeg Loss.
>
> **Q3) Question on extra regularization**
>
> Please refer to Common Response 4

---

> > ### Comment · Reviewer_E4AJ · 2023-11-23
> > **Response to the Author**
> >
> > Thanks for you extensive response. I start gain insights upon these addition experiments and explanations in the rebuttal page. However, I believe the paper need to be greatly re-written to integrate all the explanations above in a clear and concise way. Thus I am willing to increase my score to 5.

---

> > > ### Author Response · Authors · 2023-11-23
> > > **Thank you for the feedback!**
> > >
> > > We are pleased that our response has been helpful in gaining insights. Within a short time, we worked diligently to address the reviewers' concerns and made our best efforts to incorporate our response into the paper. In the final version, we will further enhance our paper by reflecting additional feedback from Reviewer E4AJ, aiming to make it more clearer and concise. Thank you once again for providing feedback

---

> ### Author Response · Authors · 2023-11-22
> **Author response 3**
>
> Dear Reviewer E4Aj,
>
> Thank you once again for providing insightful comments. We have made every effort to respond to all your comments within the limited discussion period and have completed the second revision, incorporating all reviewer’s comment into the manuscript.
>
> We want to reiterate that while our proposed AugNeg is a simple idea, its core contribution lies in identifying issues from a different perspective in the field of CSSL research (specifically, pointing out the problem of both 1) the conventional loss form and 2) the low diversity of negative samples). We have introduced a novel loss function to overcome these challenges via model-based augmentation of negative representations. Furthermore, we assert that our contributions are most evident in the successful application of AugNeg to six methods, including additional experiments conducted in response to Reviewer E4Aj. This demonstrates the versatility of our proposed idea in CSSL, involving both contrastive and non-contrastive algorithms. In particular, we believe that the successful integration of AugNeg with MoCo and BYOL, which exhibit the best performance in the CSSL setting, is remarkable. This integration (AugNeg with MoCo and BYOL) leads to the new state-of-the-art performance in various experiments. Additionally, we believe our approach holds valuable potential for application and extension to continual representation learning in different domains, such as multi-modal representation learning.
>
> As part of additional experiments, we are currently conducting supplementary experiments on AugNeg(VICReg) using ImageNet-100 and DomainNet in three CSSL scenarios. We plan to incorporate these results in the final camera ready version. Furthermore, we are in the process of extending our application experiments to other SSL algorithms, aiming to include the results in the final camera ready version also.
>
> As the discussion period is coming to a close, we would greatly appreciate it if the reviewer could review our response (and ideally the revised manuscript) and provide feedback. Thank you again.

---

### Author Response · Authors · 2023-11-17
**Common Response 1**

**Common Response 1 to the all reviewers**

We appreciate the detailed feedback provided for our paper. We have diligently addressed all comments and questions, striving to enhance the paper based on the reviewers' comments. We look forward to any additional feedback on our responses.

In the initial revision of our paper, we have:

1. Corrected overall typos and notation errors.
2. Updated a caption of Figures and Tables.
3. Included result graphs for additional experiments proposed in the manuscript and Supplementary Materials.

The modifications are indicated in magenta color. We will sequentially incorporate responses to each reviewer in the manuscript in further revisions.

Thank you for your valuable comments again.

---

### Author Response · Authors · 2023-11-17
**Common Response 2**

**Common Response 2 to the all reviewers**
* **Contributions of our paper.**

Before providing individual responses to each reviewer, we would like to summarize the contributions of our paper as follows:

1. Firstly, we not only introduced the challenges of the conventional loss function but also first highlighted the issue of a significant reduction in negative samples, in Continual Self-Supervised Learning (CSSL) (See Section 3.2).
2. Building on this, we proposed the AugNeg loss based on InfoNCE loss, leveraging additional negatives from each current and previous task model (Equation 2).
3. Additionally, we demonstrated that the idea of utilizing these additional negative samples can be extended to non-contrastive learning-based algorithms as well (See Section 3.4).
4. We conducted extensive experiments on various datasets, CSSL scenarios, and downstream tasks, demonstrating superior performance compared to the state-of-the-art method with a minimal difference in training cost (See Section 4).


We believe that **despite being based on a simple idea,** **our proposed algorithm is well-motivated (as acknowledged by reviewers Tkp5 and GiPi)**.
Also, the proposed idea is not limited to the computer vision domain, suggesting its potential applicability in a wide range of domains (e.g., Natural Language Processing, Multi-modal Learning, etc.) for continual representation learning. We would appreciate it if reviewers consider these points when making the final decision.

---

### Author Response · Authors · 2023-11-17
**Common Response 3**

**Common Response 3 to all reviewers**
* **The role and effect of using additional negatives for AugNeg**

The analysis of the gradients for the proposed AugNeg Loss is already presented in the Supplementary Materials, specifically in section A.1, titled "Analysis on the gradients." In this analysis, we analytically demonstrated how additional negatives contribute to the gradient.

Additionally, through various experiments in the manuscript, we demonstrated the superiority of training with the proposed AugNeg Loss. In the ablation study in Table 5, we compared the results by removing each loss term's additional negatives (e.g., $\mathcal{N_t}$ and $\mathcal{N_{t-1}}$) and by doubling the queue size of negative samples. These comparisons serve as evidence of the effectiveness of AugNeg.

Considering these results, **we believe that the AugNeg Loss in the CSSL clearly makes a difference in the gradient, leading to excellence in learning better representations**. Moreover, We believe that AugNeg's success in integrating with MoCo can be attributed to its ability to incorporate a significant number of negatives from both the $t$ and $t-1$ models, all without requiring a large mini-batch. Additionally, leveraging negatives queued from previous iterations serves a role similar to exemplars, helping to mitigate catastrophic forgetting. For additional experimental validation on this, please refer to Section B.7 in the Supplementary Materials.

To enhance clarity, we will provide updates in the additional revision of the manuscript.

---

### Author Response · Authors · 2023-11-17
**Common Response 4**

**Common Response 4 to all reviewers**
* **More explanation on Equation 5**

We introduced the motivation behind Equation 5 in Section 3.4 in the manuscript. For further clarification, we will employ the notation used in the manuscript and introduce the SimCLR implementation of the second term ($\mathcal{L}_2$) constituting the proposed AugNeg Loss.

When representing the $i$-th input image as $x_i$, applying distinct augmentations $A$ and $B$ to N input images results in an augmented batch $x'=[x^A_1, ..., x^A_N, x_{N+1}^B, ... x_{2N}^B]$ with 2N elements (where $i = 1, ..., N$ and $x_i = x_{i+N}$). Following the notation in the manuscript, we can define the output feature of task $t$ model as $z\_{i, t} = \text{Proj}(h\_{\theta\_t}({x}'\_i)) \in \mathcal{R}^{D_P}$ and $\tilde{z}\_{i, t} = \text{Pred}(z\_{i, t}) \in \mathcal{R}^{D_P}$. Based on this, the SimCLR implementation of the $\mathcal{L}\_2$ in AugNeg Loss is as follows:

$$\mathcal{L}\_{\text{2}}(x\_i,\theta_t,\theta\_{t-1}) = - {\mathrm{log}}\frac{\mathrm{exp}(\hat S\_{i,i} / \tau)}{\sum_{k=1}^{2N}{\mathbb{1}\_{[k\neq i]}\mathrm{exp}( S\_{i,k} / \tau) + {\sum\_{k=1}^{2N}{\mathbb{1}\_{[k\neq i]}\mathrm{exp}(\hat S\_{i,k} / \tau)}}}}$$

Here, $S\_{i,j}=(\tilde{z}\_{i, t})^\top ({z}\_{j, t}) / (||(\tilde{z}\_{i, t})^\top||\cdot ||({z}\_{j, t})||)$ and $\hat S\_{i,j}=(\tilde{z}\_{i, t})^\top ({z}\_{j, t-1}) / (||(\tilde{z}\_{i, t})^\top||\cdot ||({z}\_{j, t-1})||)$ represent the terms in the numerator and denominator, respectively. As introduced in the manuscript, this is a similar form with the CaSSLe's distillation term, incorporating the idea of considering additional negatives($\sum\_{k=1}^{2N}{\mathbb{1}{[k\neq i]}\mathrm{exp}( S\_{i,k} / \tau)}$) into the denominator. Comparing this formula to Equation (4) in Section 3.4, we can consider that the numerator part ($\mathrm{exp}(\hat S\_{i,i} / \tau)$) plays a role similar to Equation (4). However, since BYOL does not fully consider negative samples for self-supervised learning, terms like the denominator of $\mathcal{L}\_{\text{2}}$ do not exist.

The motivation behind the regularization defined in Equation (5) for considering negatives from the t-1 model stems from one of the terms in the denominator of $\mathcal{L}\_{\text{2}}$, namely $\sum\_{k=1}^{2N}{\mathbb{1}\_{[k\neq i]}\mathrm{exp}(\hat S\_{i,k} / \tau)}$. It considers all 2N-1 samples as negatives for the given augmented $x\_i$ with the condition that $k\neq i$ excluding only one sample. For instance, among the 2N-1 negatives for $x^A\_1$, an image with the different augmentation applied to the same source input $x$ (e.g., $x^B\_{1+N}$) is included as the negatives. As a result, the distance between the output features of the $t$ model for $x^A\_1$ and the $t-1$ model for $x^B\_{1+N}$ increases during training task $t$.

Equation 5 ($-|\text{Pred}(h_{\theta_t}(x^{\text{A}})) - h_{\theta_{t-1}}(x^{\text{B}})|^2_2$) is an BYOL implementation for considering the negatives mentioned above. Specifically, for images with different augmentations applied to the same source image $x$, denoted as $x^A$ and $x^B$, this additional regularization is applied to increase the squared mean square error between the output features of each $t$ and $t-1$ model to ensure that they differ. With these explanations in mind, we would like to underscore that the additional regularization introduced to incorporate negatives from the t-1 model for CSSL using non-contrastive learning is inspired by the AugNeg Loss presented in Section 3.3.

We will incorporate the above content into the future revision of the manuscript to provide a clearer understanding of the motivation behind the proposed regularization. Once again, thank you for your valuable comments.

---

### Author Response · Authors · 2023-11-22
**Common Response 5**

Dear Reviewers,

We sincerely appreciate the reviewer’s time and effort in reviewing our manuscript, and we express our gratitude for providing valuable comments. We have completed the second revision, integrating our responses to each reviewer's insightful feedback into the paper. The summarized overview of the overall revisions is as follows:

1. **General Proofreading:**
   - Replaced lengthy sentences and clarified ambiguous expressions throughout the manuscript.
   - Modified sentences to avoid overconfidence in the proposed method.
   - Improved the clarity of the key contributions in our paper.

2. **Related Work:**
   -  Integrated citations to previous works on SSL augmentation methods, addressing **Reviewer GiPi's comment (Weakness 3)**.
   - Added a paragraph highlighting the key distinctions between our paper and others.

3. **Method:**
   - Clarified that negatives used for model-based augmentation are obtained during the training of the current task $t$, addressing **Reviewer E4Aj's comment (Question 1)**.
   - Incorporated details from Common Responses 3 and 4 into the Supplementary Materials and made corresponding comments in Sections 3.3 and 3.4 to address **common feedback from all reviewers**.
   - Addressed **Reviewer E4Aj's comment (Weakness 1)** by providing a brief explanation of the implementation of AugNeg(VICReg) in Section 3.4 and moving additional details to the Supplementary Materials.

4. **Experiments:**
   - Incorporated additional results for AugNeg(VICReg) obtained in response to **Reviewer E4Aj's comment (Weakness 1)** into Figure 3 and Table 1.
   - Added a discussion on the performance improvement of AugNeg when considering Joint in Section 4.2, addressing **Reviewer Tkp5 (Weakness 1)**.
   - Modified the analysis in Section 4.4 based on **Reviewer Tkp5's comment (Weakness 3)**, and included content related to the suggested analysis of AugNeg(BYOL) in the Supplementary Materials.

5. **Concluding Remarks:**
   - Rewrote the conclusion to ensure clarity of our paper.

6. **Supplementary Materials:**
   - Added the loss functions for AugNeg(MoCo), AugNeg(SimCLR), and AugNeg(VICReg) in section A.3, providing explanations.
   - Expanded section A.4 to include details related to **Common Response 4**.
   - In section B.3, added the results of the suggested analysis for AugNeg(BYOL) from **Reviewer Tkp5's comment**.
   - Included additional experimental results in section B.5 in response to **Reviewer E4Aj's comment (Weakness 1)**.
   - Updated the discussion on different queue sizes for AugNeg(MoCo) in section B.8 to address **common feedback from all reviewers**.

We believe these revisions have substantially improved the quality and clarity of our manuscript. We hope the updated version meets the reviewers’ expectation. Thank you once again for your valuable feedback.

---

### Meta-Review · Area_Chair_RvVz · 2023-12-18

**Metareview:**

The paper proposes a modification to self-supervised based continual learning methods. Overall, it seems like a promising approach and the reviewers all agreed that the idea seemed intuitive. There were all borderline ratings with two borderline negative (5) and one borderline positive (6). Broadly, the main concerns stem from clarity of presentation (i.e. motivation of approach and description of approach) and empirical results (typos, unable to reproduce baselines etc). Upon personally reading a large part of the manuscript, I agree with the reviewers' concerns that the presentation could be significantly improved. It seems intuitive that more negative examples will help, but how exactly to incorporate them and what the two different loss functions (L1 AND L2) are and how they differ from the "conventional loss form" was insufficiently explained. Furthermore, it would be good to proofread all the experiment results carefully, and incorporate all the suggested experiments into the main paper for the next version.

**Justification For Why Not Higher Score:**

Concerns about soundness of empirical results and baselines, as well as lack of clarity in presentation of the method - raised by 2 out of the 3 reviewers and the AC agrees with the concerns.

**Justification For Why Not Lower Score:**

N/a

---

### Decision · Program_Chairs · 2024-01-16

Reject